# Powerful one-dimensional scan to detect heterotic quantitative trait loci

Guoliang Li [ID], Renate H. Schmidt [ID], Yusheng Zhao [ID], Jochen C. Reif [ID] ✉ & Yong Jiang [ID] ✉

To meet the growing global demand for food, increasing yields through heterosis in agriculture is crucial. A deep understanding of the genetic basis of heterosis has led to the development of a quantitative genetic framework that incorporates both dominance and epistatic effects. However, incorporating all pairwise epistatic interactions is computationally challenging due to the large sequencing depth and population sizes needed to uncover the genes behind complex traits. In this study, we develop hQTL-ODS, a one-dimensional scanning method that directly assesses the net contribution of each quantitative trait locus to heterosis. Simulations show that hQTL-ODS reduces computational time while offering higher power and lower false-positive rate. We apply this method to a population of 5243 wheat hybrids with whole-genome sequenced profile, revealing key epistatic hubs that play a critical role in determining heterosis.

Heterosis, the enhanced performance of offspring compared to their parents, is a phenomenon whose exploitation contributes significantly to food security[1,2]. It is systematically leveraged in both livestock[3–5] and crop plants[6–8] through hybrid breeding. Heterosis can be explained genetically either by the overdominant effect of individual genes[9–11], by the complementary combination of several genes with (partially) dominant gene action[12–14] or by specific epistatic interactions between genes[15–17]. The explanatory hypotheses are not mutually exclusive. Interestingly, the causes of heterosis in most crop and livestock species are poorly understood, but a proper theoretical framework and declining sequencing costs pave the way to fill these gaps through genetic association studies.

The heterotic effect of a gene or quantitative trait locus (QTL) can be attributed to its dominance effect and its epistatic interaction effects with other loci[18,19]. Based on the quantitative genetic description of the heterotic effect, a multi-step scanning procedure[19] was developed to detect heterotic QTL (hQTL) in a hybrid population created by crossing different parents: QTL scans are performed for individual components, i.e., dominance effects of all markers and digenic epistatic (additive-by-additive, additive-by-dominance, and dominance-by-dominance) effects of all marker pairs, followed by a test that integrates all significant components of a given putative heterotic hQTL (hQTL-MSS). This is the only existing method that considers both dominance and epistatic effects to detect hQTL in a diverse hybrid population. However, the challenge with this approach is that as the number of markers increases to for instance a million, the computational burden becomes prohibitive with trillions of digenetic epistatic interactions to be tested. This is particularly challenging in large populations which are needed to ensure high power for QTL detection. One solution to this problem is to avoid screening all individual components and test directly for the net heterotic effect of a locus, reducing the number of tests to the number of markers.

In our study, we develop a novel one-dimensional scanning (hQTL-ODS) method to test the net contribution of individual loci to heterosis (Fig. 1). This allows us to substantially reduce the computational time of genome-wide searches for heterosis loci in large populations. In simulation studies, we show that the method has higher statistical power with fewer false positives. Capitalizing on large hybrid wheat populations genotyped with whole-genome resequencing (WGS) data, we test the strategy and show the importance of the contribution of cumulative small epistatic interactions to heterosis in wheat.

Leibniz Institute of Plant Genetics and Crop Plant Research (IPK) Gatersleben, Corrensstraße 3, Seeland, Germany. ✉e-mail: reif@ipk-gatersleben.de; jiang@ipk-gatersleben.de

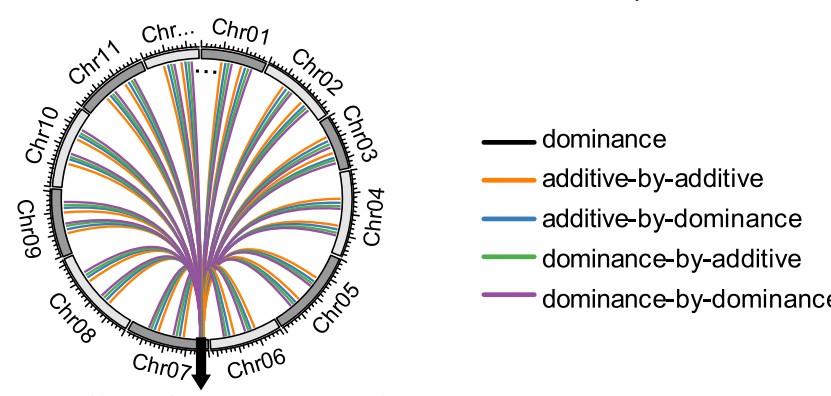

**Definition of heterotic effect for a QTL**

- dominance
- additive-by-additive
- additive-by-dominance
- dominance-by-additive
- dominance-by-dominance

Heterotic effect of the $i$-th marker for a particular hybrid $F$:

$$h_{i,F} = \begin{cases} d_i - \dfrac{1}{2}\sum_{j\in R_{20}} aa_{ij} + \dfrac{1}{2}\sum_{j\in R_{02}} aa_{ij} + \dfrac{1}{2}\sum_{j\in R_{22}} da_{ij} - \dfrac{1}{2}\sum_{j\in R_{00}} da_{ij} + \dfrac{1}{2}\sum_{j\in R_{20}\cup R_{02}} dd_{ij}, & if\ i\in R_{20} \\[2mm] d_i - \dfrac{1}{2}\sum_{j\in R_{02}} aa_{ij} + \dfrac{1}{2}\sum_{j\in R_{20}} aa_{ij} + \dfrac{1}{2}\sum_{j\in R_{22}} ad_{ji} - \dfrac{1}{2}\sum_{j\in R_{00}} da_{ij} + \dfrac{1}{2}\sum_{j\in R_{20}\cup R_{02}} dd_{ij}, & if\ i\in R_{02} \\[2mm] \dfrac{1}{2}\sum_{j\in R_{20}\cup R_{02}} ad_{ij}, & if\ i\in R_{22} \\[2mm] -\dfrac{1}{2}\sum_{j\in R_{20}\cup R_{02}} ad_{ij}, & if\ i\in R_{00} \end{cases}$$

Jiang et al. Nat. Genet., 2017

**hQTL-ODS** 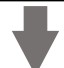 **hQTL-MSS**

**Heterotic effect of the $i$-th marker for all hybrids (reformulated in vector form):**

$$\boldsymbol{h}_i = \boldsymbol{T}\left[ \boldsymbol{l}_i d_i + \frac{1}{2}\sum_{\substack{j=1\\ j\neq i}}^{p} \left( (\boldsymbol{m}_i \circ \boldsymbol{m}_j) aa_{ij} + (\boldsymbol{m}_i \circ \boldsymbol{l}_j) ad_{ij} \right.\right.$$
$$\left.\left. + (\boldsymbol{l}_i \circ \boldsymbol{m}_j) ad_{ji} + (\boldsymbol{l}_i \circ \boldsymbol{l}_j) dd_{ij} \right) \right]$$

GWAS for dominance and epistatic effects (two-dimensional scan)

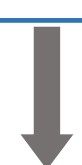

**Directly testing $\boldsymbol{h}_i$ using likelihood ratio test (one-dimensional scan)**

$$\lambda_{LR} = -2[\ln(L_0) - \ln(L_1)]$$

$L_0$: maximum likelihood of the null model
$$\boldsymbol{y}_{MPH} = \boldsymbol{X\alpha} + \boldsymbol{g}_D + \boldsymbol{g}_{AA} + \boldsymbol{g}_{AD} + \boldsymbol{g}_{DD} + \boldsymbol{\varepsilon}$$

$L_1$: maximum likelihood of the alternative model
$$\boldsymbol{y}_{MPH} = \boldsymbol{X\alpha} + \boldsymbol{h}_i + \boldsymbol{g}_D + \boldsymbol{g}_{AA} + \boldsymbol{g}_{AD} + \boldsymbol{g}_{DD} + \boldsymbol{\varepsilon}$$

$\boldsymbol{X\alpha}$: covariate effects; $\boldsymbol{g}_D$, $\boldsymbol{g}_{AA}$, $\boldsymbol{g}_{AD}$, $\boldsymbol{g}_{DD}$: genetic background Effects; $\boldsymbol{\varepsilon}$: residuals

Integrating significant component effects into heterotic effects

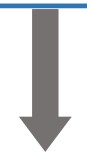

Testing heterotic effects

**Fig. 1 | An overview of hQTL-ODS in comparison to the existing approach hQTL-MSS[19].**

## Results
### Methods overview

The hQTL-ODS model directly tests the heterotic effect of a marker, avoiding extensive association tests for pairwise interaction effects (Fig. 1). For each marker, the following two linear mixed models

(LMMs) are compared:

$$\boldsymbol{y}_{MPH} = \boldsymbol{X\alpha} + \boldsymbol{g}_D + \boldsymbol{g}_{AA} + \boldsymbol{g}_{AD} + \boldsymbol{g}_{DD} + \boldsymbol{\varepsilon}, \qquad (1)$$

$$\boldsymbol{y}_{MPH} = \boldsymbol{h}_i + \boldsymbol{X\alpha} + \boldsymbol{g}_D + \boldsymbol{g}_{AA} + \boldsymbol{g}_{AD} + \boldsymbol{g}_{DD} + \boldsymbol{\varepsilon}, \qquad (2)$$

where $\mathbf{y}_{MPH}$ is the vector of mid-parent heterosis (MPH) values for all hybrids. Model (1) is the null model containing covariate effects ($\mathbf{X}\boldsymbol{\alpha}$), multiple genetic background effects ($\mathbf{g}_D$, $\mathbf{g}_{AA}$, $\mathbf{g}_{AD}$, $\mathbf{g}_{DD}$) and the residuals ($\boldsymbol{\varepsilon}$), but excluding any marker effects. The alternative model (2) includes the heterotic effect ($\mathbf{h}_i$) of the tested marker in addition to all effects in the null model. The heterotic effect $\mathbf{h}_i$ is defined as a complex linear combination of ($4p-3$) effects, i.e. the dominance effect $d_i$ and the epistatic effects $aa_{ij}, ad_{ij}, da_{ij}, dd_{ij}$ ($j = 1, \ldots, p$ and $j \neq i$, where $p$ is the number of markers)[19]. In the model, $\mathbf{h}_i$ is treated as a random vector following a multi-variate normal distribution: $\mathbf{h}_i \sim N(0, \mathbf{H}_i \sigma_i^2)$, where $\mathbf{H}_i$ is a covariance matrix and $\sigma_i^2$ is a variance component. A key step to implement the model is to derive an alternative expression of $\mathbf{h}_i$ which is less complex than (but equivalent to) the original definition. It is also crucial to find an efficient method of calculating $\mathbf{H}_i$ (see **Methods** for more details). After solving the model by the restricted maximum likelihood (REML) method, the significance of $\mathbf{h}_i$ is assessed by testing the null hypothesis $\sigma_i^2 = 0$ using the likelihood ratio test. Namely, the test statistic is of the form $LR_i = -2\ln(L_0/L_1)$, where $L_0$ is the maximum likelihood of the null model, and $L_1$ is the maximum likelihood of the alternative model. Under the null hypothesis, this test statistic follows approximately a mixture of $\chi_0^2$ and $\chi_1^2$ distributions with equal weights[20,21], where $\chi_0^2$ and $\chi_1^2$ refer to chi-squared distributions with zero and one degree of freedom.

## hQTL-ODS is computationally more efficient than hQTL-MSS

The new model hQTL-ODS has a clear advantage in computational time complexity, compared with hQTL-MSS. In hQTL-MSS, GWAS has to be performed for dominance effects of all markers and for digenic epistatic effects of all marker pairs based on an LMM similar to model (2), replacing $\mathbf{h}_i$ by these component effects. Thanks to the P3D approximation[22] it is sufficient to solve the null model (1) once. Since the model contains multiple random vectors ($\mathbf{g}_D$, $\mathbf{g}_{AA}$, $\mathbf{g}_{AD}$, $\mathbf{g}_{DD}$) in addition to the residuals, the computational load cannot be reduced by applying eigen-decomposition to the covariance matrices, a commonly implemented technique for LMMs with a single random vector. Thus, the time complexity of this step is $O(tn^3)$, where $n$ is the number of individuals and $t$ is the number of iterations, which depends on the numerical method used to solve the model and is less than 10 in many cases. Computing the test statistics for dominance and epistatic effects takes $O(n^2p)$ and $O(n^2p^2)$ time, respectively. In case of WGS data, it is expected that $p \gg tn$. Thus, $n^2p^2 \gg n^2p \gg tn^3$, indicating that the time complexity of hQTL-MSS is dominated by the procedure of testing digenic epistatic effects, namely $O(n^2p^2)$. In hQTL-ODS, it is necessary to solve $p+1$ LMMs (the null model once and the alternative model for each marker), resulting in $O(tn^3p)$ time. The time for producing the likelihood ratio test statistics is merely $O(p)$, which is negligible. Additional time is needed for hQTL-ODS to calculate the covariance matrix $\mathbf{H}_i$ for each marker. By implementing an efficient algorithm for calculating genomic epistatic relationship matrices[23], the time required for calculating $\mathbf{H}_i$ for all markers can be reduced from $O(n^2p^2)$ to $O(n^2p)$ (see "Methods" for details). Hence, the time complexity of hQTL-ODS is dominated by solving $p+1$ LMMs, namely $O(tn^3p)$. Considering $p \gg tn$, we have $n^2p^2 \gg tn^3p$. Therefore, the hQTL-ODS model is computationally much more efficient than hQTL-MSS. More precisely, the time required by hQTL-MSS is approximately $\frac{n^2p^2}{tn^3p} = \frac{p}{tn}$ times as long as that required by hQTL-ODS. For example, if $t \approx 10$, hQTL-ODS is about 100 times faster than hQTL-MSS for a data set with 1000 individuals and one million markers.

The theoretical advantage of hQTL-ODS mentioned above was validated by using a wheat data set consisting of 1557 hybrids with 1.2 million single-nucleotide polymorphisms (SNPs). The data set was termed Exp I (Supplementary Data 1, see "Methods" for details). From this data set, we sampled subsets with four marker numbers (5000, 10,000, 20,000 and 50,000) in combination with four sample sizes (200, 500, 1000, and 1557). hQTL-ODS and hQTL-MSS were performed

for each subset on a computing platform with 50 CPU cores (Intel(R) Xeon(R) Gold 6130 CPU @ 2.10 GHz) and 24 GB memory size per CPU. We observed that the running time of hQTL-ODS ($t_{ODS}$) was substantially shorter than that of hQTL-MSS ($t_{MSS}$), and the relative advantage measured as the ratio $\rho = t_{MSS}/t_{ODS}$ depended on the population size and the number of markers (Fig. 2a). When the population size is fixed, the ratio linearly increased relative to the number of markers, which is well in accordance with the theory that $\rho$ approximately equals $p/tn$ (Fig. 2b–e). In the entire population with 1557 genotypes, the observed trend of $\rho$ values almost completely overlapped with the theoretical line of t = 5 (Fig. 2e). This means that hQTL-ODS would be about 154 times faster than hQTL-MSS for the full data set (1557 genotypes with all 1.2 million markers). While it took 180 h for hQTL-ODS to finish the analysis on the full data set, it would take more than three years for hQTL-MSS to complete the task.

## Statistical power and false-positive rate

hQTL-ODS is also theoretically superior to hQTL-MSS in terms of properly modeling heterotic effects. In hQTL-ODS, the heterotic effects were modeled unbiasedly according to the original definition, i.e., the dominance effect of a locus and its epistatic interaction effect with all other loci were considered, independent of their effect sizes. In contrast, hQTL-MSS filtered the component effects by testing them and applying a threshold, hence only modeled large effects. When many small component effects cumulatively make a large contribution, the heterotic effect estimated by hQTL-MSS is likely to be biased.

To verify the above hypothesis, we compared the performance of hQTL-ODS and hQTL-MSS in a simulation study. MPH values were simulated following model (2) based on the genomic data of a subset of Exp I, consisting of 1000 hybrids with 5000 SNPs. Five scenarios were considered in order to take different patterns of component genetic effects composing hQTL into account (Supplementary Data 2, see "Methods" for details). In all scenarios, only one hQTL was simulated and the proportion of phenotypic variance explained (PVE) by the hQTL was fixed at 2.5%. Despite the small PVE, the simulated hQTL in all scenarios were adequately captured by hQTL-ODS, with a detection power ranging from 45 to 63%, and a false positive rate (FPR) of less than 0.07% (Fig. 2f, g). When a few epistatic effects cumulatively contributed to the effect of the simulated hQTL, no matter whether the dominance effect contributed (Scenarios 3 and 4) or not (Scenarios 1 and 2), hQTL-ODS consistently outperformed hQTL-MSS, with higher power and lower FPR (Fig. 2f, g). Only in Scenario 5, where the hQTL effect consisted of the dominance effect of the QTL alone, the power of both methods was the same. These results clearly validated the theoretical superiority of hQTL-ODS over hQTL-MSS.

We also investigated the association between the hQTL detection power and heterozygosity (Supplementary Data 3). A positive association was observed in all scenarios in which dominance effects played a role (Scenarios 3, 4, and 5), however, it was only significant when epistatic effects did not contribute at all (Scenario 5). The association was weaker still when only a few epistatic effects shaped out the hQTL (Scenario 2). It became negative and non-significant when many epistatic effects contributed to the hQTL effect (Scenario 1). These results are in line with expectations. When the dominance effect is the main component of the hQTL effect, the detection power is naturally associated with heterozygosity. However, if cumulative epistatic effects play an important role, the picture becomes much more complex because many factors, such as the type of epistasis (additive-by-additive, additive-by-dominance, or dominance-by-dominance), the number of loci interacting with the hQTL and the sizes of epistatic effects, could affect the detection power.

## A large resequenced hybrid wheat panel

In order to test the performance of hQTL-ODS on experimental data, three hybrid wheat populations, each from independent experimental

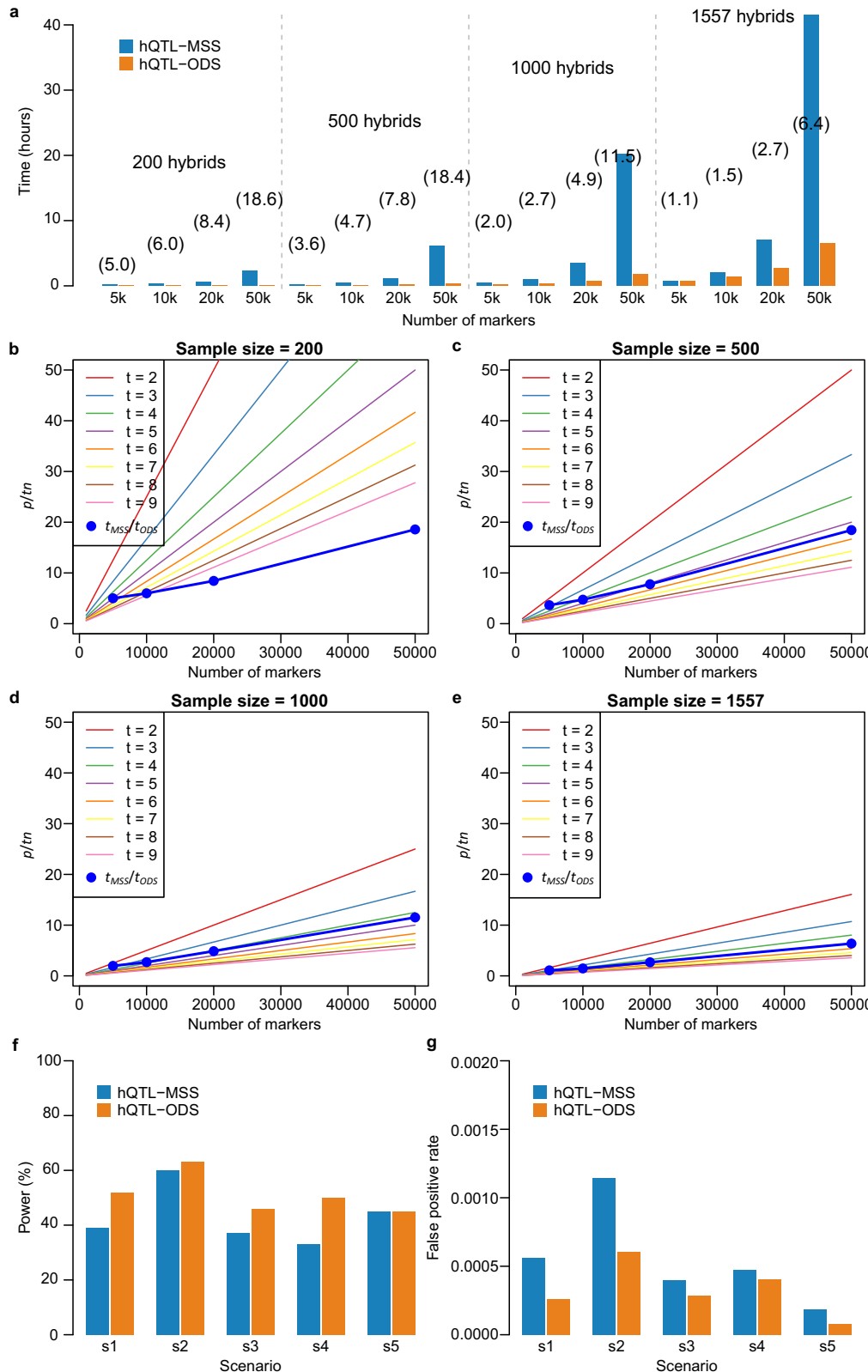

**Fig. 2 | Comparison of running time, power, and false-positive rate between hQTL-ODS and hQTL-MSS on experimental and simulated data sets. a** The running time of hQTL-ODS ($t_{ODS}$) and hQTL-MSS ($t_{MSS}$) on experimental data sets with different sample sizes and marker numbers. The numbers in parentheses represent the ratio $\rho = t_{MSS}/t_{ODS}$. **b–e** The relative advantage of hQTL-ODS over hQTL-MSS. Theoretically, hQTL-ODS is approximately $p/tn$ times faster than hQTL-MSS, where $p$ is the number of markers, $n$ is the sample size, and $t$ is the average number of iterations for solving the linear mixed model. The lines depicted the

expected values $p/tn$ and the blue dots indicated the observed ratio $\rho = t_{MSS}/t_{ODS}$ on experimental data sets. **f** The statistical power and **g** The false-positive rate of hQTL-ODS and hQTL-MSS evaluated on simulated data sets for five different scenarios (s1-s5). The simulated hQTL effect was contributed by epistatic effects (s1 and s2), by both dominance and epistatic effects (s3 and s4), or by dominance effects (s5). More details on the simulation scenarios were described in "Methods" and Supplementary Data 2).

series (Exp I, II, and III)[24], were assembled into an integrated panel consisting of 5243 hybrids derived from 597 parental lines representing a comprehensive selection of the Central European elite bread-wheat breeding pool. In each Exp, hybrids and parental lines were evaluated in field trials for grain yield and heading date in at least 10 year-by-location combinations (Supplementary Data 1, 4).

The parental lines were resequenced using WGS technology, and sufficient read coverage was established for 588 genotypes (Supplementary Data 5). Read mapping and SNP calling against the reference genome assembly of cv. Chinese Spring v2.1 (RefSeq CSv2.1)[25] provided 7,835,467 SNPs with minor allele frequency (MAF) larger than or equal to 0.05 and with an average density of 0.54 SNPs per kb. SNPs whose pairwise linkage disequilibrium (LD, estimated by $r^2$) was higher than 0.9 were pruned within 50 kb sliding window and 907,534 high-quality SNPs remained for subsequent analysis. The genotypes of the hybrids were inferred from the corresponding parental lines. Combining the genomic and phenotypic data resulted in 4,885 hybrids derived from 545 parental lines.

A principal coordinate analysis (PCoA) revealed that neither the parental lines from each population nor across populations grouped into clusters (Supplementary Fig. 1a). Different Exps were linked by up to 26 common genotypes showing high trait correlations between different Exps (Supplementary Data 6), suggesting that integration of the data did not introduce a systematic bias. Analysis in the integrated panel showed that the LD decayed on average to at least half of its maximum value at a distance of 10 Mb (Supplementary Fig. 1b). High broad-sense heritability estimates were observed for grain yield and heading date performance of hybrids (0.77, 0.94) and parents (0.88, 0.98) (Supplementary Fig. 1c, d, Supplementary Data 4). The MPH of grain yield varied widely from −0.86 to 2.02 Mg/ha and had a mean value of 0.81 Mg/ha, whereas for heading date the MPH ranged between −3.90 and 3.05 days with a mean value of −0.93 days (Supplementary Fig. 1e, f, Supplementary Data 4). The heritability estimates of MPH were 0.68 and 0.78 for grain yield and heading date, respectively (Supplementary Fig. 1c, d, Supplementary Data 7). Thus, the data provide a reliable foundation to dissect the genetic architecture of heterosis for the traits of interest.

**Resequencing data uncover the advantages of hQTL-ODS**
Before exploring hQTL in the integrated data set, we compared the performance of hQTL-ODS with hQTL-MSS in Exp I using a low-density marker panel. In a previous study, hQTL-MSS had been applied to Exp I genotyped by a 90k Illumina Infinium SNP chip to study the MPH of grain yield[19]. Thus, we directly adopted the results therein (Fig. 3a) and applied hQTL-ODS to the same data set (Fig. 3b). hQTL-MSS and hQTL-ODS detected 37 and 19 hQTL, respectively, of which only nine and eight colocalized with each other (Fig. 3d). A certain degree of inconsistency was not unexpected, as it has been shown both in theory and in simulation that hQTL-MSS might result in a biased estimation of the heterotic effect due to ignoring small genetic component effects.

As mentioned in the previous subsection, the population was also genotyped by WGS. A comparison of the SNPs obtained via WGS with those of the 90k SNP chip revealed an average SNP concordance rate of 95.6%, indicating that results of subsequent analyses based on the two marker panels can be reliably compared. Then, hQTL-ODS was applied to this population with WGS data and 165 hQTL were identified, 124 of which mapped to known chromosomes (Fig. 3c). Comparing with the results obtained using the SNP chip, we found that 105 hQTL were only detected with the WGS data (Fig. 3d), which demonstrated the superiority of high-density marker panels. Notably, 15 of the 105 hQTL (13.9%) showed neither significant dominance nor epistatic interactions effects with other loci, indicating that cumulative small epistatic effects composed these hQTL. They can only be detected using hQTL-ODS as it does not ignore small component

effects contributing to the heterotic effect. Interestingly, we observed that nine hQTL identified by hQTL-ODS using the SNP chip were not detected by the same model with WGS data. Furthermore, six hQTL detected by hQTL-MSS using the SNP chip colocalized with 11 hQTL detected by hQTL-ODS when adopting WGS, but missed by hQTL-ODS using the SNP chip (Fig. 3d). Thus, using a low-density marker panel, the estimation of heterotic effect by hQTL-ODS may be biased because not all genetic variants are considered. The most reliable results are consequently expected by applying hQTL-ODS with a high-density marker panel, such as WGS.

**hQTL-ODS revealed pervasive cumulative epistatic effects**
We applied hQTL-ODS to the integrated panel as well as to the three individual populations (Exp I, II and III) using WGS data to dissect the genetic architecture of heterosis for grain yield and heading date. Thus, results were obtained in four different populations (Fig. 4a–d, Supplementary Fig. 2a–d). For convenience, hQTL detected in at least two populations were called common hQTL, and those identified in only one population were termed unique hQTL. For grain yield, 3174 significant SNPs (including those without assigned chromosome information) merged into 188 hQTL were detected in the integrated population (Supplementary Data 8), with a high proportion (74.5%) being common hQTL (Fig. 4f). In contrast, proportions of common hQTL identified in the three individual populations only amounted to 37.6% (Exp I), 39.7% (Exp II) and 25% (Exp III). Similar results were obtained for the trait heading date (Supplementary Fig. 2a–d, f). Thus, the reliability of hQTL detection was clearly enhanced in the integrated population because of its large sample size.

From now on, we focused exclusively on the results obtained with the integrated panel. The PVE of the 188 hQTL for grain yield ranged from 1.67 to 13.59%. The hQTL with the highest PVE was detected on chromosome 6B (Integrated_GY_hQTL139), spanning a large region from 47.65 Mb to 124.41 Mb and encompassing 595 significant SNPs. The two most significant hQTL (i.e., containing SNPs with lowest $p$ values) were detected on chromosome 7B (Integrated_GY_hQTL163) and 4 A (Integrated_GY_hQTL128), with 11.18% and 12.62% PVE, respectively. GWAS for dominance effects contributing to heterosis revealed only two dominance QTL (dQTL) on chromosomes 5B and 7B (Fig. 4e, Supplementary Data 9), both colocalized with hQTL. On the other hand, many significant epistatic interactions between hQTL and other loci in the genetic background were detected (Fig. 4g–j). For heading date, 71 hQTL were found with PVE ranging from 2.42 to 17.22% (Supplementary Data 10). Notably, *Ppd-D1* (TraesCS2D03G0156800)[26], which colocalized with a dQTL, represents a candidate gene for an hQTL with 8.86% PVE. In total, five dQTL contributing to heterosis were found and three colocalized with hQTL (Supplementary Fig. 2e, Supplementary Data 11). Thus, for both traits, a large number of hQTL contrasted with few QTL for dominance effects, demonstrating the overriding importance of epistatic interactions for hQTL in wheat.

Next, we exemplarily investigated the peak SNPs of three hQTL for grain yield which did not colocalize with any dQTL in more detail to disentangle the composition of component effects contributing to these hQTL effects. Indeed, the dominance effects of their peak SNPs were not significant (Fig. 5a–c). Thus, the hQTL effects must be comprised of epistatic effects, as shown in the distribution of phenotypic values for two-locus genotype classes consisting of the peak SNP and the SNP which has the strongest interaction with it (Fig. 5d–f). Interestingly, the peak SNPs of the three hQTL had distinct patterns of cumulative epistatic effects with other SNPs. SNP_4A_748345003 in Integrated_GY_hQTL128 had many significant ($P < 0.05$ after Bonferroni correction) epistatic interactions with other SNPs (Fig. 5g), whereas only several significant epistatic effects were identified for SNP_6B_653527460 in Integrated_GY_hQTL142 (Fig. 5h). Importantly, SNP_6B_180998007 in Integrated_GY_hQTL149 had no significant

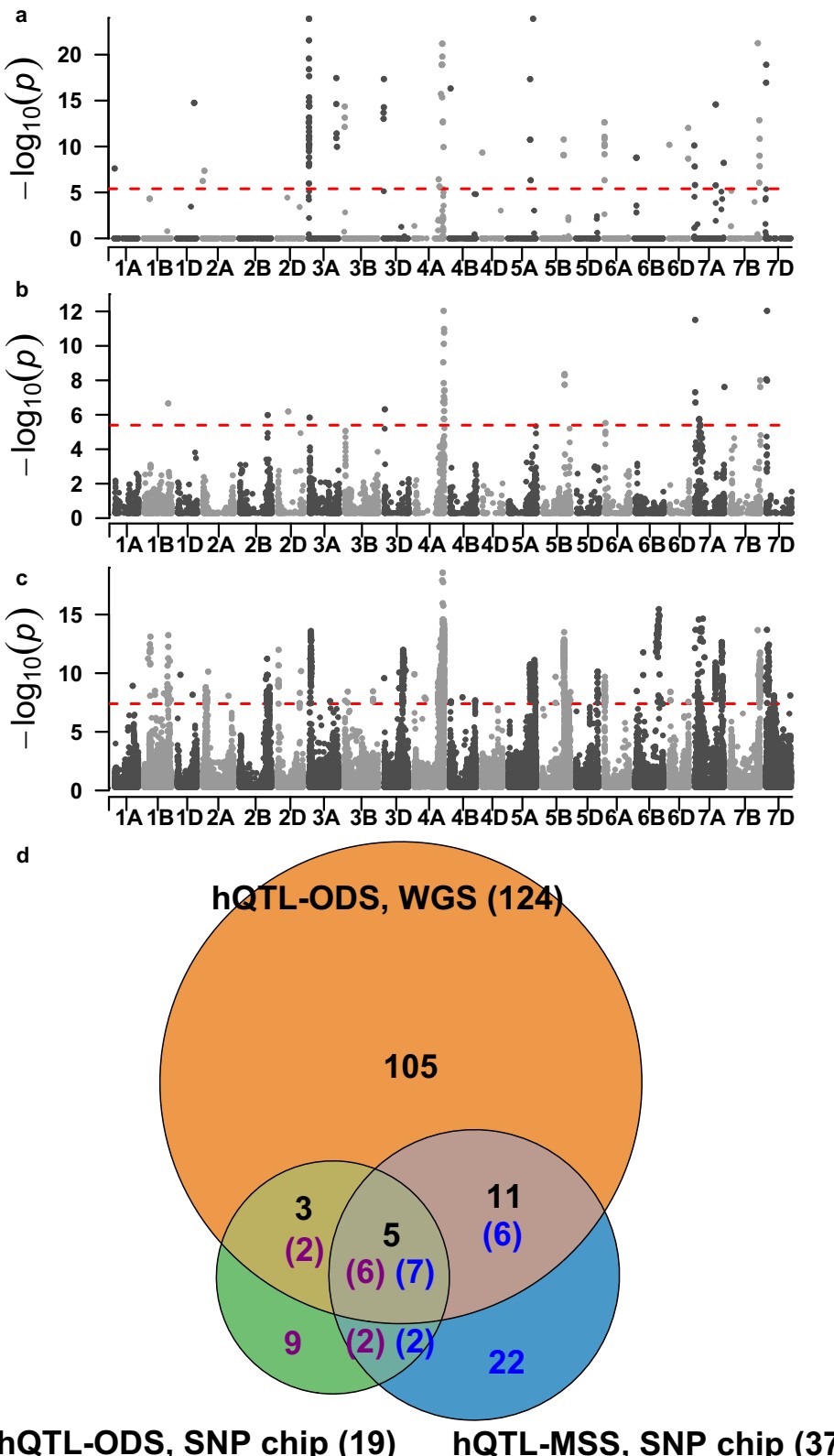

**Fig. 3 | A comparison of hQTL for grain yield heterosis detected by hQTL-ODS and hQTL-MSS using experimental series I data.** Manhattan plots showing the results of (**a**) hQTL-MSS using 90k SNP chip data (17,372 SNPs), which were adopted from Jiang et al. [19], **b** hQTL-ODS using the same 90k SNP chip, and (**c**) hQTL-ODS using whole-genome sequencing data (about 1.2 million SNPs). In (**a**–**c**), significance was assessed by the likelihood ratio test, and the thresholds ($P < 0.05$ after Bonferroni-Holm correction for multiple testing) were indicated as red dashed horizontal lines. **d** Venn diagram showing the number of hQTL detected in (**a**), (**b**), and (**c**), (indicated in blue, purple, and black, respectively) as well as the over-lapping. Note that it is possible that different numbers of hQTL detected in two or three data sets colocalized, since significant SNPs were merged into hQTL in each of the three data sets separately. To be consistent with the results from Jiang et al. [19], we only considered hQTL located on known chromosomes.

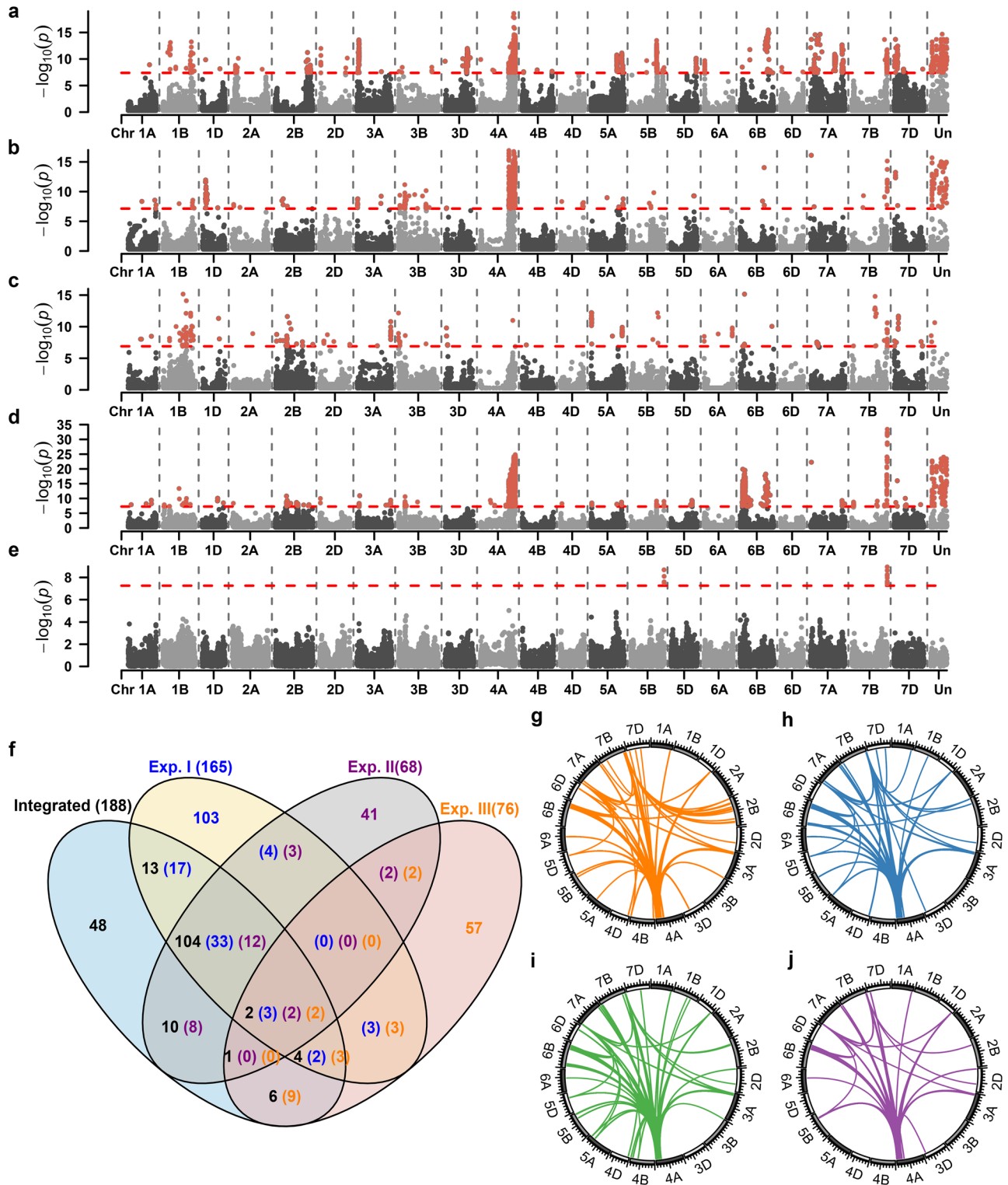

**Fig. 4 | Genetic architecture of midparent heterosis (MPH) for grain yield in wheat.** Results of hQTL-ODS revealed genomic regions harboring loci associated with grain yield MPH based on whole-genome sequencing data in the experimental series I (**a**), II (**b**), III (**c**), and the integrated population (**d**). **e** Results of a genome-wide scan for dominance effects of grain yield MPH in the integrated population. In (**a**–**e**), significance was assessed by the likelihood ratio test, and the thresholds ($P < 0.05$ after Bonferroni-Holm correction for multiple testing) were indicated as red dashed horizontal lines. **f** Venn diagram showing the number of hQTL detected in (**a**), (**b**), (**c**) and (**d**) (indicated in blue, purple, orange, and black, respectively). Note that since the procedure of merging significant SNPs into hQTL was

performed in each of the four data sets separately, an hQTL detected in one data set may colocalize with multiple hQTL detected in another data set. Therefore, the number of overlapping hQTL was separately indicated for each data set. The significant digenic epistatic interactions between hQTL and all other SNPs were shown by the colored links in the centers of the circles for additive-by-additive (**g**), additive-by-dominance (**h**), dominance-by-additive (**i**), and dominance-by-dominance interactions (**j**). In (**g**–**j**), significance was assessed by the Wald test, and the thresholds were determined as $P < 0.01$ after Bonferroni-Holm correction for multiple testing.

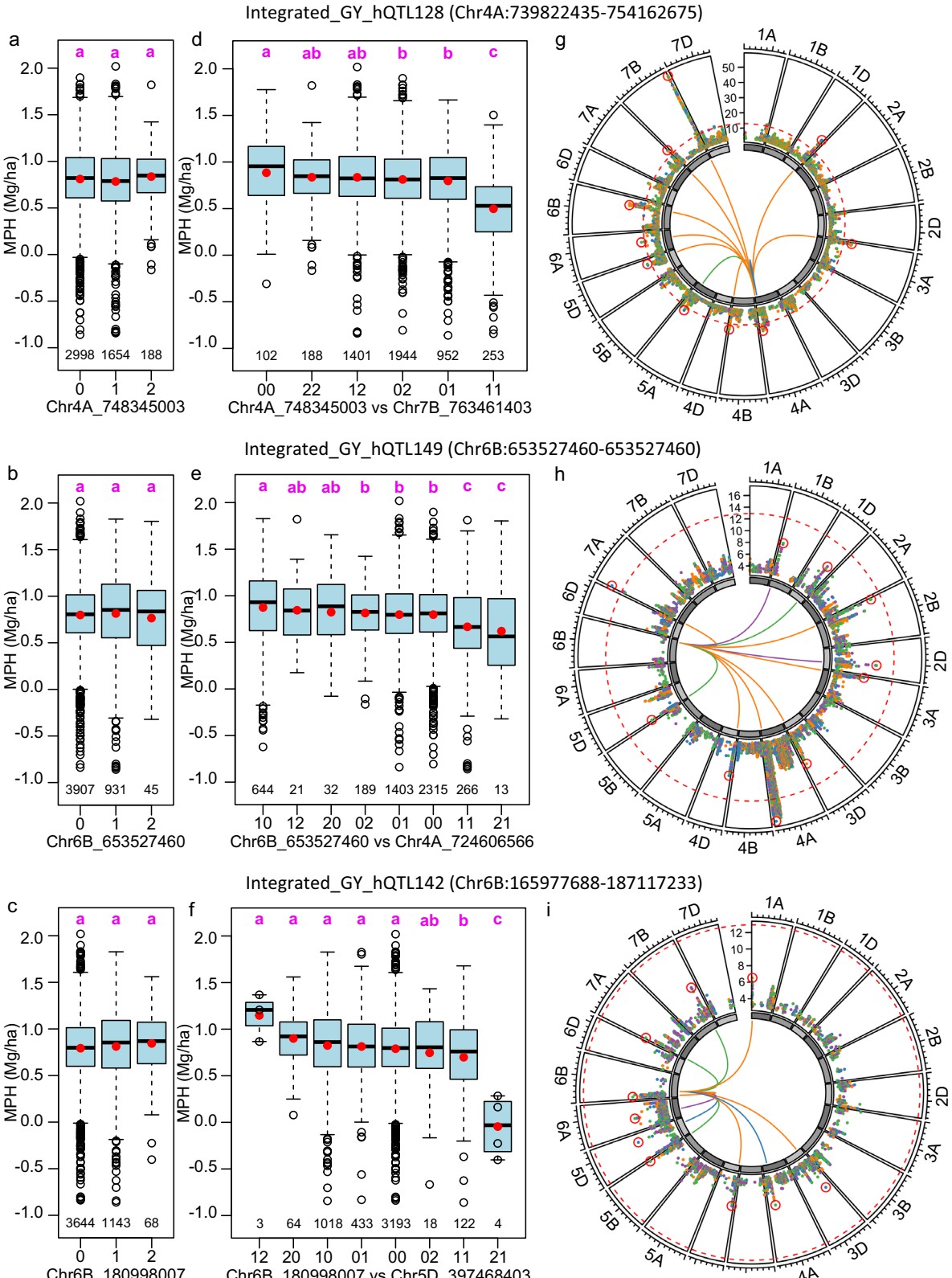

interactions with any other SNP (Fig. 5i). These latter results indicated that significant epistatic effects are not necessary for a marker showing a significant heterotic effect. Instead, small epistatic effects can cumulatively compose an hQTL. SNP_6B_180998007 is a representative for 66 out of 188 hQTL (35.1%) that can only be detected using hQTL-ODS as it does not ignore small component effects contributing to the heterotic effect.

### Testing hQTL-ODS with a large maize dataset

To further test the applicability and potential of our hQTL-ODS model, we analyzed a published maize dataset[8]. The population consists of 6210 single-cross hybrids from 207 maternal lines and 30 paternal testers. All parental lines were genotyped via whole-genome rese-quencing, and 699,441 SNPs were retained for subsequent analysis. Phenotypic data are available for three traits: days to tasseling (DTT),

**Fig. 5 | Distinct patterns of cumulative epistatic effects highlighted by three heterotic QTL (hQTL).** The three hQTL were detected for grain yield MPH in the integrated data set: Integrated_GY_hQTL128 (**a**, **d**, **g**), Integrated_GY_hQTL149 (**b**, **e**, **h**), and Integrated_GY_hQTL142 (**c**, **f**, **i**). **a**–**c** The phenotypic distribution of grain yield MPH for hybrids harboring different alleles for the peak SNP within hQTL. **d**–**f** The phenotypic distribution of grain yield MPH for hybrids with distinct genotype combinations of the peak SNP within hQTL and the SNP showing the most significant epistatic interaction effect with the peak SNP. In each group, the mean MPH value was indicated by red dots and the number of hybrids was shown below the boxplot. Boxes indicated the interquartile range (IQR, the difference between the 75th and the 25th percentile) with the median line, whiskers extend to the most

extreme values within $1.5 \times$ IQR. Different letters above the boxes indicated significant differences among groups as determined by the Least Significant Difference (LSD) test ($P < 0.05$), while groups sharing the same letter are not significantly different. **g**–**i** Circled Manhattan plot of epistatic effects between the peak SNP of hQTL and all other SNPs in which the 10 SNPs with most significant epistatic effect was highlighted by red circles and linking lines in the center. Different colors were used to indicate the types of epistatic effects: orange (additive-by-additive), blue (additive-by-dominance), green (dominance-by-additive) and purple (dominance-by-dominance). Significance was assessed by the Wald test, and the thresholds ($P < 0.05$ after Bonferroni-Holm correction for multiple testing) were indicated as red dashed lines.

plant height (PH), and ear weight (EW). We focused on EW. Using hQTL-ODS, we identified one hQTL on chromosome 3 (Supplementary Fig. 3a) with the peak SNP chr3.s_157746554 ($P = 2.56 \times 10^{-8}$). This SNP maps within a distance of 1.28 Mb to *ZmMADS69*. *ZmMADS69* functions as a flowering activator but due to pleiotropy also affects other traits of agronomic importance such as ear size[27]. Xiao et al. [8] had identified QTL for the trait DTT spanning *ZmMADS69* and also detected QTL for EW at positions near *ZmMADS69* using a different GWAS approach but with a mild significance threshold. In addition, merged QTL very close to *ZmMADS69* were listed for EW with comparatively large epistatic effects. Thus, our results are consistent with the original study's findings. Notably, no significant dominance effects were detected (Supplementary Fig. 3b), which again demonstrated the importance of considering cumulative epistatic effects when detecting hQTL.

## Discussion

In this study, we developed hQTL-ODS, a reliable and time-efficient association analysis tool that comprehensively incorporates both dominance and epistatic effects to unravel the genetic architecture of heterosis, making it feasible to conduct GWAS for heterotic effects using large-scale diverse hybrid populations with WGS data. Previous studies on heterosis in various species (e.g., Arabidopsis[28,29], rice[1,7,30] and maize[8]) were based on WGS data with diverse hybrid populations. However, epistatic effects were either not considered or restricted to the interaction between dominance QTL in these studies, possibly due to the high computational burden. It was shown that considering epistasis is important for studying heterosis in wheat[19,31], maize[32] and pigeonpea[33]. These studies applied the hQTL-MSS approach to test epistatic effects between all marker pairs, either based on a low-density marker panel or based on WGS in a relatively small population. As demonstrated in Results, it would be infeasible to apply hQTL-MSS to large populations with WGS. Our new model hQTL-ODS successfully removed the computational bottleneck by modeling the heterotic effect as a whole, hence avoiding the two-dimensional scan for digenic epistatic effects between all pairs of markers. Importantly, the computational efficiency is not the only advantage of hQTL-ODS over existing approaches. Treating a heterotic effect in its entirety averts the possible bias of focusing only on large component effects, as the heterotic effect of a locus can be significant due to the cumulation of small interaction effects between this locus and many other loci (Fig. 5i). Thus, even if the computational load of hQTL-MSS could possibly be reduced by integrating more efficient GWAS algorithms for the two-dimensional epistasis scan (e.g. REMMAX[34], NGG[35]), hQTL-ODS would still be superior because of higher power and fewer false positives (Fig. 2f, g).

It is important to note that the heterotic effect $\mathbf{h}_i$ was treated as a random vector with a zero mean in hQTL-ODS. Consequently, the mean heterosis in the model is zero. However, this seems mis-specified for traits with non-zero mean heterosis, such as grain yield in our wheat datasets. While it is common to assume that all genetic background effects have a zero mean[36], the effect under test in a GWAS model is usually treated as a fixed covariate[37]. If we keep this setting, the mean heterosis will not be zero. However, $\mathbf{h}_i$ is a combination of (4p-3)

genetic component effects and the number of components exceeds the number of observations, rendering it difficult to perform a proper statistical test. Therefore, we changed the assumption to make the likelihood ratio test feasible. The results of our simulation study in which a non-zero dominance effect was simulated (Scenarios 3, 4 and 5, Fig. 2f, g) indicated that hQTL-ODS is robust for data sets with non-zero mean heterosis values. An interesting alternative approach would be to use a directional dominance model, in which the dominance effects have non-zero mean values. This is equivalent to including a covariate for the overall heterozygosity[38]. Similarly, it may be possible to consider epistatic effects with non-zero means. This could be incorporated into the further development of the hQTL-ODS model.

From the general viewpoint of statistical genetics, the framework of hQTL-ODS is a kernel-based association test for a set of genetic variants[39,40]. Thus, although the model presented here is specific to the study of MPH, its framework can be easily expanded to various other applications. For instance, it can be used to study the better-parent heterosis (BPH) instead of MPH, provided that the matrix which transforms the vector of original trait values to MPH is appropriately modified. It can also be applied to study original traits, for which one only needs to omit the step of applying the transformation matrix (see Supplementary Notes for details). In this regard, our model is similar to the so-called marginal epistasis test implemented in MAPIT[41] and FAME[42]. The difference is that in these two models only the cumulative additive-by-additive effects were considered, whereas all three types of digenic epistatic effects were integrated in hQTL-ODS. Since the model transformation from the original trait to MPH is only implemented in hQTL-ODS but not in the marginal epistasis tests, only the former model can be applied to study heterosis.

Although hQTL-ODS delivers a substantial improvement compared with existing approaches in detecting heterotic QTL, there are still limitations in its computational efficiency if one considers the ever-growing data size. The theoretical time complexity of hQTL-ODS is about $O(tn^3p)$, indicating that the computing time will increase by a factor of 27 if the sample size triples. As a result, it took about 30 days to complete the analysis for the integrated panel with 4885 hybrids and 907,534 SNPs, while it only took hours for the three individual populations (each with about 1600 hybrids). Thus, in future, it is necessary to explore possibilities to further increase the computational efficiency. One option might be to switch from the LR test to the score test, possibly at the cost of decreasing power[39]. An even more important direction is to increase the efficiency of solving LMMs, which is the most time-consuming step in hQTL-ODS. Currently, we utilized a fast REML approach implemented in the R package gaston[43]. Note that the model involves multiple random vectors of genetic effects with dense covariance matrices. For this type of LMM, only a few fast algorithms are available: the Min-Max algorithm implemented in MM4LMM[44], the Method-of-Moments approach in GEMMA[45] and RHE-mc[46], the Monte Carlo REML method in BOLT-REML[47], and a recently developed fast REML algorithm called MPH[48]. It would be worthwhile to investigate whether the computational efficiency of hQTL-ODS can be further improved by exploiting and/or combining some of the techniques implemented in these algorithms, or whether it

may even be necessary to develop new efficient algorithms for the future.

## Methods

### MPH and the heterotic effect

In this study, we consider a population consisting of $n$ hybrids derived from crossing $r$ diverse parental lines. For a trait in consideration, the MPH of a hybrid is defined as the difference between the trait values of the hybrid and the average of its parents. Let $\mathbf{y}_{\mathrm{ori}}$ be the $(n+r)$-dimensional vector of the observed original trait values of all hybrids and parents, and $\mathbf{y}_{\mathrm{MPH}}$ be the $n$-dimensional vector of the MPH values of all hybrids. Then, it is clear from the definition that $\mathbf{y}_{\mathrm{MPH}} = \mathbf{T}\mathbf{y}_{\mathrm{ori}}$, where $\mathbf{T}$ is an $n \times (n+r)$ matrix of linear transformation.

The heterotic effect of a marker is the net contribution of this particular marker to MPH, taking its dominance effect and its digenic epistatic interaction with the entire genetic background into consideration. In particular, it depends not only on the genotype of the hybrid, but also on the genotype combinations of the parental lines. In the following, we briefly recall its precise definition[19].

Assume that all markers are biallelic and let $p$ be the number of all markers. The genotype of an individual at a marker is coded as 0, 1, or 2, depending on the number of reference alleles. Let $F$ be a hybrid individual, $P_1$ and $P_2$ be its parental lines. Let $d_i$ be the dominance effect of the $i$-th marker, $aa_{ij}$, $ad_{ij}$, $da_{ij}$ (or equivalently, $ad_{ji}$) and $dd_{ij}$ be the additive-by-additive, additive-by-dominance, dominance-by-additive and dominance-by-dominance epistatic effect between the $i$-th and the $j$-th marker, respectively. Let $R_{kl}$ ($k$, $l = 0$ or $2$) be the subset of markers for which $P_1$ has genotype code $k$ and $P_2$ has code $l$. Then, for any $i$ ($1 \le i \le p$), the heterotic effect of the $i$-th marker for this particular hybrid individual $F$, denoted by $h_{i,F}$, is defined as follows:

$$h_{i,F} = \begin{cases} d_i - \frac{1}{2}\sum\limits_{j \in R_{20}} aa_{ij} + \frac{1}{2}\sum\limits_{j \in R_{02}} aa_{ij} + \frac{1}{2}\sum\limits_{j \in R_{22}} ad_{ji} - \frac{1}{2}\sum\limits_{j \in R_{00}} ad_{ji} + \frac{1}{2}\sum\limits_{j \in R_{20} \cup R_{02}} dd_{ij} & \text{if } i \in R_{20} \\ d_i - \frac{1}{2}\sum\limits_{j \in R_{02}} aa_{ij} + \frac{1}{2}\sum\limits_{j \in R_{20}} aa_{ij} + \frac{1}{2}\sum\limits_{j \in R_{22}} ad_{ji} - \frac{1}{2}\sum\limits_{j \in R_{00}} ad_{ji} + \frac{1}{2}\sum\limits_{j \in R_{20} \cup R_{02}} dd_{ij} & \text{if } i \in R_{02} \\ \frac{1}{2}\sum\limits_{j \in R_{20} \cup R_{02}} ad_{ij} & \text{if } i \in R_{22} \\ -\frac{1}{2}\sum\limits_{j \in R_{20} \cup R_{02}} ad_{ij} & \text{if } i \in R_{00} \end{cases}. \quad (3)$$

With the above definition, the MPH value of the individual $F$ is the sum of the heterotic effects of all markers, i.e., $y_{MPH,F} = \sum_{i=1}^{p} h_{i,F}$.

Let $\mathbf{h}_i$ be the $n$-dimensional vector consisting of the heterotic effects $h_{i,F}$ for all hybrids. Let $\mathbf{M}_A$ be the $(n+r) \times p$ matrix of markers coded as 1 (homozygous for the reference allele), 0 (heterozygous) or −1 (homozygous for the alternative allele), and $\mathbf{M}_D$ be the $(n+r) \times p$ matrix of markers coded as 0 (homozygous) or 1 (heterozygous). Then, $\mathbf{h}_i$ can be written as follows:

$$\mathbf{h}_i = \mathbf{T}\left[ \mathbf{l}_i d_i + \frac{1}{2}\sum\limits_{\substack{j=1 \\ j \ne i}}^{p} \left( (\mathbf{m}_i \circ \mathbf{m}_j) aa_{ij} + (\mathbf{m}_i \circ \mathbf{l}_j) ad_{ij} + (\mathbf{l}_i \circ \mathbf{m}_j) ad_{ji} + (\mathbf{l}_i \circ \mathbf{l}_j) dd_{ij} \right) \right], \quad (4)$$

where $\mathbf{m}_i$ and $\mathbf{m}_j$ are the $i$-th and the $j$-th columns of the matrix $\mathbf{M}_A$, $\mathbf{l}_i$ and $\mathbf{l}_j$ are the $i$-th and the $j$-th columns of $\mathbf{M}_D$, and "$\circ$" denotes entry-wise product of two vectors. Let $\boldsymbol{\gamma}_i$ be the $(4p-3)$-dimensional vector consisting of $d_i$, $aa_{ij}$, $ad_{ij}$, $ad_{ji}$ and $dd_{ij}$, and $\mathbf{Z}_i$ be the $n \times (4p-3)$ matrix whose columns consist of $\mathbf{Tl}_i$, $\frac{1}{2}\mathbf{T}(\mathbf{m}_i \circ \mathbf{m}_j)$, $\frac{1}{2}\mathbf{T}(\mathbf{m}_i \circ \mathbf{l}_j)$, $\frac{1}{2}\mathbf{T}(\mathbf{l}_i \circ \mathbf{m}_j)$ and $\frac{1}{2}\mathbf{T}(\mathbf{l}_i \circ \mathbf{l}_j)$, for all $1 \le j \le p$ and $j \ne i$. Then, Eq. (4) can be rewritten in a compact form as $\mathbf{h}_i = \mathbf{Z}_i \boldsymbol{\gamma}_i$.

Note that it is not straightforward to obtain Eq. (4) from Eq. (3), as the expression of $h_{i,F}$ in terms of the genetic component effects ($d_i$, $aa_{ij}$, $ad_{ij}$, $ad_{ji}$ and $dd_{ij}$) depends on the combination of parental genotypes (i.e., whether $i \in R_{20}$, $R_{02}$, $R_{22}$ or $R_{00}$). A proof is provided in the Supplementary Notes.

### The hQTL-ODS model

Since MPH is a derived trait, it is necessary to model the genetic effects in a way that they are consistent to those contributing to the original trait. Thus, the baseline model is the following model for the original trait, taking the additive and dominance effects of all markers, and digenic epistatic effects of all marker pairs into account:

$$\mathbf{y}_{\mathrm{ori}} = \mathbf{1}_{n+r}\mu + \mathbf{X}_c\boldsymbol{\alpha} + \mathbf{M}_A\mathbf{a} + \mathbf{M}_D\mathbf{d} + \mathbf{M}_{AA}\mathbf{aa} + \mathbf{M}_{AD}\mathbf{ad} + \mathbf{M}_{DD}\mathbf{dd} + \mathbf{e}, \quad (5)$$

where $\mathbf{y}_{\mathrm{ori}}$, $\mathbf{M}_A$ and $\mathbf{M}_D$ were defined in the previous subsection (recall that the columns of $\mathbf{M}_A$ and $\mathbf{M}_D$ are denoted by $\mathbf{m}_i$ and $\mathbf{l}_i$ ($1 \le i \le p$), respectively), $\mathbf{1}_{n+r}$ is the $(n+r)$-dimensional vector of ones, $\mu$ is the common intercept term, $\boldsymbol{\alpha}$ is a $k$-dimensional vector of covariate effects (e.g., subpopulation effects), $\mathbf{X}_c$ is the corresponding $(n+r) \times k$ design matrix, $\mathbf{a}$ and $\mathbf{d}$ are $p$-dimensional vector of coded genotypic effects[49] for all markers, $\mathbf{aa}$, $\mathbf{ad}$ and $\mathbf{dd}$ are vectors of epistatic interaction effects between $\mathbf{a}$ and $\mathbf{d}$ for all marker pairs, $\mathbf{aa}$ and $\mathbf{dd}$ are $p(p-1)/2$-dimensional, $\mathbf{ad}$ is $p(p-1)$-dimensional. $\mathbf{M}_{AA}$ is an $(n+r) \times p(p-1)/2$ matrix whose columns consist of $\mathbf{m}_i \circ \mathbf{m}_j$ for all $i,j$ such that $1 \le i < j \le p$. $\mathbf{M}_{AD}$ is an $(n+r) \times p(p-1)$ matrix whose columns consist of $\mathbf{m}_i \circ \mathbf{l}_j$ for all $i,j$ such that $1 \le i,j \le p$ and $i \ne j$. $\mathbf{M}_{DD}$ is an $(n+r) \times p(p-1)/2$ matrix whose columns consist of $\mathbf{l}_i \circ \mathbf{l}_j$ for all $i,j$ such that $1 \le i < j \le p$, $\mathbf{e}$ is an $(n+r)$-dimensional vector of residuals.

Since $\mathbf{y}_{\mathrm{MPH}} = \mathbf{T}\mathbf{y}_{\mathrm{ori}}$, the model for MPH is naturally obtained by left-multiplying the matrix $\mathbf{T}$ to both sides of Eq. (5):

$$\mathbf{y}_{\mathrm{MPH}} = \mathbf{T}\mathbf{1}_{n+r}\mu + \mathbf{T}\mathbf{X}_c\boldsymbol{\alpha} + \mathbf{T}\mathbf{M}_A\mathbf{a} + \mathbf{T}\mathbf{M}_D\mathbf{d} + \mathbf{T}\mathbf{M}_{AA}\mathbf{aa} + \mathbf{T}\mathbf{M}_{AD}\mathbf{ad} + \mathbf{T}\mathbf{M}_{DD}\mathbf{dd} + \mathbf{T}\mathbf{e}. \quad (6)$$

The definition of $\mathbf{T}$ implies that $\mathbf{T}\mathbf{1}_{n+r} = \mathbf{0}_n$. For each hybrid, the marker coding of $\mathbf{M}_A$ is exactly the average of the coding of its parents, implying $\mathbf{T}\mathbf{M}_A = \mathbf{0}_{n \times p}$. Let $\mathbf{X} = \mathbf{T}\mathbf{X}_c$, $\mathbf{g}_D = \mathbf{T}\mathbf{M}_D\mathbf{d}$, $\mathbf{g}_{AA} = \mathbf{T}\mathbf{M}_{AA}\mathbf{aa}$, $\mathbf{g}_{AD} = \mathbf{T}\mathbf{M}_{AD}\mathbf{ad}$, $\mathbf{g}_{DD} = \mathbf{T}\mathbf{M}_{DD}\mathbf{dd}$ and $\boldsymbol{\varepsilon} = \mathbf{T}\mathbf{e}$. Then, Eq. (6) can be written as the following:

$$\mathbf{y}_{\mathrm{MPH}} = \mathbf{X}\boldsymbol{\alpha} + \mathbf{g}_D + \mathbf{g}_{AA} + \mathbf{g}_{AD} + \mathbf{g}_{DD} + \boldsymbol{\varepsilon}, \quad (7)$$

which is the same as Eq. (1) in the main text. In the model, $\boldsymbol{\alpha}$ is assumed to be fixed effects, $\mathbf{g}_D \sim N(0, \mathbf{K}_D\sigma_D^2)$, $\mathbf{g}_{AA} \sim N(0, \mathbf{K}_{AA}\sigma_{AA}^2)$, $\mathbf{g}_{AD} \sim N(0, \mathbf{K}_{AD}\sigma_{AD}^2)$, $\mathbf{g}_{DD} \sim N(0, \mathbf{K}_{DD}\sigma_{DD}^2)$, and $\boldsymbol{\varepsilon} \sim N(0, \mathbf{TT}'\sigma_\varepsilon^2)$. Note that the covariance matrix for $\boldsymbol{\varepsilon}$ is $\mathbf{TT}'$ instead of an identity matrix, because the residuals $\mathbf{e}$ of the original trait are usually assumed to be independent (i.e., $\mathbf{e} \sim N(0, \mathbf{I}_{n+r}\sigma_\varepsilon^2)$) and $\boldsymbol{\varepsilon} = \mathbf{T}\mathbf{e}$.

To calculate the covariance matrices $\mathbf{K}_*$ (* denotes D, AA, AD or DD), it would be natural to deduce them from Eq. (6) as $\mathbf{K}_* = \mathbf{T}\mathbf{M}_*\mathbf{M}_*'\mathbf{T}/c_*$, $c_* = \frac{1}{n}\mathrm{tr}(\mathbf{T}\mathbf{M}_*\mathbf{M}_*'\mathbf{T})$. In our case, a slightly different form was used such that the estimated variance components match the classical concept of dominance and epistatic variance[50], i.e., the variance of dominance and epistatic deviation from the breeding value[51]. More precisely, $\mathbf{K}_* = \mathbf{T}\mathbf{U}_*\mathbf{U}_*'\mathbf{T}/c_*$, $c_* = \frac{1}{n}\mathrm{tr}(\mathbf{T}\mathbf{U}_*\mathbf{U}_*'\mathbf{T})$, where $\mathbf{U}_*$ are matrices with the same size as $\mathbf{M}_*$. For the $i$-th genotype and the $k$-th marker, the $(i,k)$-entry in $\mathbf{U}_A$ and $\mathbf{U}_D$ are defined as follows:

$$(\mathbf{U}_A)_{i,k} = \begin{cases} 0 - 2p_k, & \text{if } (\mathbf{M}_A)_{i,k} = 0 \\ 1 - 2p_k, & \text{if } (\mathbf{M}_A)_{i,k} = 1 \\ 2 - 2p_k, & \text{if } (\mathbf{M}_A)_{i,k} = 2 \end{cases};$$

$$(\mathbf{U}_D)_{i,k} = \begin{cases} -2p_k^2, & \text{if } (\mathbf{M}_A)_{i,k} = 0 \\ 2p_k(1-p_k), & \text{if } (\mathbf{M}_A)_{i,k} = 1 \\ -2(1-p_k)^2, & \text{if } (\mathbf{M}_A)_{i,k} = 2 \end{cases};$$

where $p_k$ is the frequency of the reference allele, and $\mathbf{U}_{AA}, \mathbf{U}_{AD}, \mathbf{U}_{DD}$ are constructed by multiplying columns of $\mathbf{U}_A$ and $\mathbf{U}_D$ in the same way as deriving $\mathbf{M}_{AA}, \mathbf{M}_{AD}, \mathbf{M}_{DD}$ from $\mathbf{M}_A$ and $\mathbf{M}_D$.

Equation (7) is the null model of hQTL-ODS, the role of random genetic effects $\mathbf{g}_D, \mathbf{g}_{AA}, \mathbf{g}_{AD}$ and $\mathbf{g}_{DD}$ is for controlling the genetic relatedness in the population. If there is strong population structure, additional covariates can be included in $\mathbf{X}\boldsymbol{\alpha}$. To test the heterotic effect of a marker, the heterotic effect $\mathbf{h}_i$ is added into the model, resulting in the following alternative model:

$$\mathbf{y}_{MPH} = \mathbf{X}\boldsymbol{\alpha} + \mathbf{h}_i + \mathbf{g}_D + \mathbf{g}_{AA} + \mathbf{g}_{AD} + \mathbf{g}_{DD} + \boldsymbol{\varepsilon}, \quad (8)$$

where $\mathbf{h}_i$ has been defined in Eq. (4).

Recall that $\mathbf{h}_i = \mathbf{Z}_i \boldsymbol{\gamma}_i$, where $\boldsymbol{\gamma}_i$ is the $(4p - 3)$-dimensional vector consisting of $d_i$, $aa_{ij}$, $ad_{ij}$, $ad_{ji}$ and $dd_{ij}$ for all $1 \leq j \leq p$ and $j \neq i$. The component effects in $\boldsymbol{\gamma}_i$ were implicitly assumed to have different variance component in Eq. (7), as they are involved in the genetic background effects $\mathbf{g}_D$, $\mathbf{g}_{AA}$, $\mathbf{g}_{AD}$ and $\mathbf{g}_{DD}$. Here, in order to apply the likelihood ratio test and to avoid over-parametrization, we make an additional assumption that the component effects in $\boldsymbol{\gamma}_i$ are identically distributed. More specifically, $\boldsymbol{\gamma}_i \sim N(\mathbf{0}, \frac{1}{c_i}\mathbf{I}_n \sigma_{h_i}^2)$, where $\mathbf{I}_n$ is the $n \times n$ identity matrix, $c_i = \frac{1}{n}\mathrm{tr}(\mathbf{Z}_i \mathbf{Z}_i')$. Then, we have $\mathbf{h}_i N(\mathbf{0}, \mathbf{H}_i \sigma_{h_i}^2)$, where $\mathbf{H}_i = \mathbf{Z}_i \mathbf{Z}_i' / c_i$. As indicated by the results of the simulation study, it is unlikely that this additional assumption has a large influence on the effectiveness of our model (Fig. 2f, g).

With the above consideration, a classical likelihood ratio test is used to assess the significance of $\mathbf{h}_i$. Namely, the null model (Eq.(7)) and the alternative model (Eq.(8)) are solved by the REML approach, and the null hypothesis $H_0 : \sigma_{h_i}^2 = 0$ is tested by the following test statistic:

$$\lambda_{LR} = -2\left[\ln(L_0) - \ln(L_1)\right], \quad (9)$$

where $L_0$ and $L_1$ are the maximum values of the likelihood function of the null and the alternative model, respectively. Under the null hypothesis, $\lambda_{LR}$ follows approximately an equal-weighted mixture of two chi-squared distributions with zero ($\chi_0^2$) and one ($\chi_1^2$) degrees of freedom[20,21], i.e., $\lambda_{LR} \sim \frac{1}{2}\chi_0^2 + \frac{1}{2}\chi_1^2$.

The models (7) and (8) are implemented by using the R package gaston[43]. Other details on the implementation are provided in the Supplementary Note. Here, we just mention three important points: (1) Unlike the model of GWAS for the original traits, the residual term of hQTL-ODS model has a non-trivial covariance matrix ($\mathbf{TT}'$). Thus, a further transformation of the model has to be made before applying standard algorithms to solve the LMM. (2) Calculating the matrices $\mathbf{K}_{AA}$, $\mathbf{K}_{AD}$, $\mathbf{K}_{DD}$ and $\mathbf{H}_i$ is time-consuming if they are directly calculated from definition. Taking $\mathbf{H}_i = \mathbf{Z}_i \mathbf{Z}_i'$ as an example, it takes $O(n^2 p)$ time because the dimension of $\mathbf{Z}_i$ is $n \times (4p - 3)$. Thus, the total time complexity is $O(n^2 p^2)$ for calculating $\mathbf{H}_i$ for all $p$ markers. However, using the techniques similar to an efficient algorithm for calculating epistatic genomic relationship matrices[23], the time complexity was eventually reduced to $O(n^2 p)$. (3) The computational efficiency is further improved by fixing the ratio of the variance components $\sigma_D^2/\sigma_\varepsilon^2$, $\sigma_{AA}^2/\sigma_\varepsilon^2$, $\sigma_{AD}^2/\sigma_\varepsilon^2$ and $\sigma_{DD}^2/\sigma_\varepsilon^2$, i.e., estimating these ratios only once in the null model and assuming that they are invariant in the alternative model for each marker. This approach is similar to the so-called P3D approximation in GWAS[22].

### The model for testing genetic component effects

The hQTL-ODS model provides a direct test for the heterotic effect of a marker without testing any component effect (i.e., the dominance effect of the marker and the digenic epistatic interaction effects between the marker and the entire genetic background). However, if a marker has significant heterotic effect $\mathbf{h}_i$, it would be interesting to investigate whether these component effects, namely $d_i$, $aa_{ij}$, $ad_{ij}$, $ad_{ji}$

and $dd_{ij}$ for all $j \neq i$, $1 \leq j \leq p$, are significant or not. The following models are used for testing these effects.

$$\mathbf{y}_{MPH} = \mathbf{X}\boldsymbol{\alpha} + \mathbf{T}\mathbf{l}_i d_i + \mathbf{g}_D + \mathbf{g}_{AA} + \mathbf{g}_{AD} + \mathbf{g}_{DD} + \boldsymbol{\varepsilon}, \quad (10)$$

$$\mathbf{y}_{MPH} = \mathbf{X}\boldsymbol{\alpha} + \mathbf{T}\left(\mathbf{m}_i \circ \mathbf{m}_j\right) aa_{ij} + \mathbf{g}_D + \mathbf{g}_{AA} + \mathbf{g}_{AD} + \mathbf{g}_{DD} + \boldsymbol{\varepsilon}, \quad (11)$$

$$\mathbf{y}_{MPH} = \mathbf{X}\boldsymbol{\alpha} + \mathbf{T}\mathbf{l}_j d_j + \mathbf{T}\left(\mathbf{m}_i \circ \mathbf{l}_j\right) ad_{ij} + \mathbf{g}_D + \mathbf{g}_{AA} + \mathbf{g}_{AD} + \mathbf{g}_{DD} + \boldsymbol{\varepsilon}, \quad (12)$$

$$\mathbf{y}_{MPH} = \mathbf{X}\boldsymbol{\alpha} + \mathbf{T}\mathbf{l}_i d_i + \mathbf{T}\left(\mathbf{l}_i \circ \mathbf{m}_j\right) ad_{ji} + \mathbf{g}_D + \mathbf{g}_{AA} + \mathbf{g}_{AD} + \mathbf{g}_{DD} + \boldsymbol{\varepsilon}, \quad (13)$$

$$\mathbf{y}_{MPH} = \mathbf{X}\boldsymbol{\alpha} + \mathbf{T}\mathbf{l}_i d_i + \mathbf{T}\mathbf{l}_j d_j + \mathbf{T}\left(\mathbf{l}_i \circ \mathbf{l}_j\right) dd_{ij} + \mathbf{g}_D + \mathbf{g}_{AA} + \mathbf{g}_{AD} + \mathbf{g}_{DD} + \boldsymbol{\varepsilon}, \quad (14)$$

where all notations have been defined in the previous two subsections. In these models, the effects $d_i$, $d_j$, $aa_{ij}$, $ad_{ij}$, $ad_{ji}$, and $dd_{ij}$ are assumed to be fixed instead of random. Note that $a_i$ and $a_j$ are not included in models (11), (12), and (13), which seems to be a violation to the principle that the corresponding main effects should be included in the model when an interaction effect is tested. But actually, $\mathbf{TM}_A = \mathbf{O}_{n \times p}$ (see the previous subsection) implies $\mathbf{Tm}_i a_i = \mathbf{Tm}_j a_j = \mathbf{O}_n$. Therefore, it makes no difference whether to include these effects or not.

Let $x$ be any of the effects $d_i$, $aa_{ij}$, $ad_{ij}$, $ad_{ji}$ and $dd_{ij}$. A Wald test is used to assess its significance, and the test statistic has the following form:

$$W = \frac{\hat{x}^2}{\mathrm{var}(\hat{x})}, \quad (15)$$

where $\hat{x}$ is the estimated value of $x$ and $\mathrm{var}(\hat{x})$ is the estimated variance of $\hat{x}$. It is known that $W$ follows asymptotically a chi-squared distribution with one degree of freedom ($W \sim \chi_1^2$).

### Hybrid wheat populations and field trials

In this study, we integrated experimental data of three large hybrid wheat populations (Exp I, II, and III)[24] into a large panel. In total, there were 597 elite parental lines and 5243 $F_1$ hybrid progenies. The parental lines were chosen to reflect a wide range of diversity in Central Europe. A factorial or partial factorial design was used to generate single-cross hybrids in each population. Exp I was composed by 135 parental lines (120 females and 15 males) and their 1604 hybrid progenies evaluated for grain yield (Mg ha⁻¹) in 11 environments (five locations in 2012 and six in 2013) and heading date (days from January 1st) in 10 environments (six locations in 2012 and four in 2013) in Germany. Exp II was comprised of 226 parental lines (185 females and 41 males) and their 1815 hybrid progenies evaluated for grain yield in 12 environments (six locations each in 2016 and 2017) and heading date in 20 environments (11 locations in 2016 and nine in 2017) in Germany. Exp III consisted of 236 parental (196 female and 40 male) lines and their 1824 hybrid progenies evaluated for grain yield in 12 environments (six sites each in 2018 and 2019) and heading date in 18 environments (10 sites in 2018 and eight in 2019) in Germany. In each environment, the experimental design consisted of three trials, in which a partially replicated (Exp I) or un-replicated (Exp II and III) alpha lattice design was used. Different genotypes were evaluated in different trials linked by 10 (Exp I) or 11 (Exp II and III) common checks within the environment. For all genotypes, harvesting was performed mechanically, and the grain yield was adjusted to a moisture content of 140 g $H_2O$ kg⁻¹. A summary of the three populations was provided in Supplementary Data 1. Details of the field

trials and part of the phenotypic data for grain yield (Exp I, Exp II, and the 2018 data of Exp III) have been described in previous studies[24].

## DNA Isolation and genome sequencing

For DNA extraction and sequencing, the seeds of all parental lines in the three populations were collected and grown in the greenhouse and a single leaf per genotype was harvested from a 10-day-old seedling. Genomic DNA was isolated from young leaf tissue of each genotype using a silica-membrane technology (NucleoSpin 96 Plant II) as described by the manufacturer (Macherey-Nagel), which were used as input for sequencing library generation. Subsequently, the WGS libraries were prepared using the Nextera DNA Flex Library Prep according to the manufacturer's (Illumina) instructions. Libraries were pooled in an equimolar manner. The multiplexed pool was quantified by quantitative PCR and sequenced (paired-end, $2 \times 151$ cycles and 10 bp for the index reads) on NovaSeq 6000 at threefold coverage.

## WGS read processing and SNP calling

To avoid bias resulting from base-calling duplicates and adapter contamination, raw read sequences were processed to remove adaptors and low-quality reads with a minimum read quality (q) cutoff of 20. High-quality data with an average of about 17 Gb per sample were retained for subsequent analysis. The paired-end reads (172.8 billion) of all genotypes were aligned against the wheat reference genome (Chinese Spring, RefSeq v2.1) using the MEM algorithm of BWA with default parameters. The output was converted to binary alignment map (BAM) format file using SAMtools (v.1.9) and further sorted with NovoSort (v.3.06.05). After merging BAM files for samples sequenced on the same flowcell with Picard (v.2.21.9), variant calling was done using the mpileup and call functions from SAMtools with parameter '-DV'. Subsequently, a raw population genotype VCF file was generated, including 181,421,099 SNPs with an average density of 12.45 SNPs per kb. Bi-allelic SNPs with a minimum QUAL (i.e., mapping quality) score of 40, read depth for homozygous ≥1 and heterozygous calls ≥2 were recalled based on read depth ratios calculated from the DP (total read depth) and DV (depth of the alternative allele) fields in VCF file using a custom AWK script. The SNPs with missing rate ≤30%, heterozygosity ≤1% and counts of both homozygous genotypes ≥10 were kept. Then, the missing values were imputed by Beagle (v 5.2). Consequently, a total of 7,835,467 SNPs with MAF ≥ 5% were retained for the integrated panel (including all parental lines from Exp I, II, and III). Pairs of SNPs with squared correlation greater than 0.90 were greedily pruned within 50 kb sliding window using plink (v 1.90b6.9) with steps = 5 SNPs until no such pairs remained. The same data cleaning and filtering process was performed separately within each population (Supplementary Fig. 4). The genotypes of the hybrids were inferred from the parental genotypes.

## Population structure, LD decay and duplicate genotypes

Principal coordinate analysis (PCoA) was performed based on pairwise Rogers' distances among genotypes[52] to investigate the population structure. The decay of pairwise LD between SNPs (measured as $r^2$) was analyzed by PopLDdecay[53] with parameters "-MaxDist 50000 -OutType 2". The average physical distance at which the LD between SNPs decayed to half maximum value was about 10 Mb.

Duplicate genotypes might occur when different labels were used for the same genotype in different populations. Thus, we investigated whether duplicate parental lines existed across populations by applying a principle of genetic distance[24]. Groups of genotypes with pairwise Rogers's distances below 0.03 were defined as duplicates and were considered as the same genotype in subsequent analysis. As a result, 60 parental lines were merged into 28 groups, and 330 hybrids were accordingly merged into 162 groups.

## Curation of phenotypic data

Firstly, all data were screened for data-entry errors and outliers by using the Bonferroni-Holm test to judge the residuals standardized by rescaled median absolute deviation[54]. Then, a two-step approach was used to analyze the phenotypic data for the integrated panel. In the first step, the phenotypic data within each environment was analyzed by fitting a linear mixed model including the effect of genotypes, trials, replications nested within trials, and blocks nested within trials and replications. Best linear unbiased estimations (BLUEs) of the genotypes in each environment were obtained and served as the input of the second step, in which the following linear mixed model was fitted:

$$y_{ijk} = t_i (\mu_L + G_{L,i}) + (1 - t_i)(\mu_H + G_{H,i}) + s_j + E_k + e_{ijk}, \quad (16)$$

where $y_{ijk}$ is the BLUE of the $i$-th genotype in the $j$-th population (Exp I, II or III) evaluated in the $k$-th environment, $t_i$ is a dummy variable which equals 1 if the genotype is a parental line, and 0 if it is a hybrid, $\mu_L$ is the mean of all lines, $\mu_H$ is the mean of all hybrids, the effect of the $i$-th genotype is denoted by $G_{L,i}$ (if it is a line) or $G_{H,i}$ (if it is a hybrid), $s_j$ is the effect of the $j$-th population, $E_k$ is the effect of the $k$-th environment, and $e_{ijk}$ is the residual term. All effects except $\mu_L$, $\mu_H$ and $s_j$ were assumed to be random. For any $i$ and $j$, we assume that $G_{L,i} \sim N(0, \sigma_L^2)$, $G_{H,i} \sim N(0, \sigma_H^2)$, $E_j \sim N(0, \sigma_E^2)$, and $e_{ijk} \sim N(0, \sigma_{e,k}^2)$, where $\sigma_L^2$, $\sigma_H^2$ and $\sigma_E^2$ are the variance components of the lines, hybrids and residuals, respectively, and $\sigma_{e,k}^2$ is the residual variance in the $k$-th environment (i.e., we assume variance heterogeneity for the residuals in different environments). Covariance between each pair of these variables was assumed to be zero.

The significance of each variance component was tested by the likelihood ratio test. The broad-sense heritability for the lines and hybrids was calculated separately using the following formulas:

$$H_{\text{line}}^2 = \frac{\sigma_L^2}{\sigma_L^2 + \sigma_e^2/N_{E,L}}, \quad H_{\text{hybrid}}^2 = \frac{\sigma_H^2}{\sigma_H^2 + \sigma_e^2/N_{E,H}}, \quad (17)$$

where $\sigma_e^2$ is the average residual variance across all environments, $N_{E,L}$ and $N_{E,H}$ are the average numbers of environments in which the lines and hybrids were evaluated, respectively.

Then, model (16) was fitted once again with slightly different assumptions, namely $G_{L,i}$ and $G_{H,i}$ were assumed to be fixed effects instead of random, to obtain the cross-environment BLUEs of the genotypic values for all hybrids and parental lines.

Similar analysis was performed separately within each population. In this case, the model (16) was fitted without the effect $s_j$.

## Estimating the heritability of MPH

For the integrated panel, we first calculated the MPH value for each hybrid within each environment based on the within-environment BLUEs of all hybrids and parental lines. Then, the following linear mixed model was fitted:

$$y_{\text{MPH},ijk} = \mu_{\text{MPH}} + G_{\text{MPH},i} + s_{\text{MPH},j} + E_{\text{MPH},k} + \varepsilon_{ijk}, \quad (18)$$

where $y_{\text{MPH},ijk}$ is the MPH value of the $i$-th hybrid of the $j$-th population in the $k$-th environment, $\mu_{\text{MPH}}$ is the mean across populations, $G_{\text{MPH},i}$ is the genotypic effect of the $i$-th hybrid, $s_{\text{MPH},j}$ is the effect of the $j$-th population, $E_{\text{MPH},k}$ is the effect of the $k$-th environment, and $\varepsilon_{ijk}$ is the residual. In the model, $\mu_{\text{MPH}}$ and $s_{\text{MPH},j}$ are assumed to be fixed and the other effects are random: $G_{\text{MPH},i} \sim N(0, \sigma_{G,\text{MPH}}^2)$, $E_{\text{MPH},k} \sim N(0, \sigma_{E,\text{MPH}}^2)$, and $\varepsilon_{ijk} \sim N(0, \sigma_{\varepsilon,k}^2)$, where $\sigma_{G,\text{MPH}}^2$ and $\sigma_{E,\text{MPH}}^2$ are the variance components of genotypes and environments, $\sigma_{\varepsilon,k}^2$ is the residual variance in the $k$-th environment.

Then, the broad-sense heritability of MPH was estimated using the following formula:

$$H^2_{\text{MPH}} = \frac{\sigma^2_{\text{G, MPH}}}{\sigma^2_{\text{G, MPH}} + \sigma^2_\varepsilon / N_{\text{E, H}}}, \tag{19}$$

where $\sigma^2_\varepsilon$ is the average residual variance across all environments and $N_{\text{E, H}}$ is the average number of environments in which the hybrids were evaluated, as in Eq. (17).

Similar analysis was performed separately within each population. In this case, the model (18) was fitted without the effect $s_{\text{MPH},j}$ to estimate the within-population heritability of MPH.

All linear mixed models in this and the previous subsection were implemented using the package ASReml-R 4.1[55].

## Applying hQTL-ODS to the hybrid wheat data set

For each population (Exp I, II, and III) as well as the integrated panel, the genomic and phenotypic data were combined for detecting hQTL by using the hQTL-ODS model. The final number of parental lines, hybrids and markers are shown in Supplementary Fig. 4. In particular, there were 4885 hybrids and 545 parental lines with 907,534 high-quality SNPs in the integrated panel. Since no clear population stratification was observed (Supplementary Fig. 1), we didn't include any covariate in the model (i.e., setting $\mathbf{X\alpha} = \mathbf{0}$ in Eqs.(7) and (8). $P$ values obtained in the LR test for heterotic effects were corrected for multiple testing with the Bonferroni-Holm method[56]. The genome-wide threshold was determined to be $P < 0.05$ after correction. Manhattan plots were generated using the R package CMplot[57] (version 4.5.1), and circular plots were generated using the R package circlize[58] (version 0.4.13).

After the genome-wide scan of heterotic effects, the following approach was used to merge significant SNPs into hQTL. First, an interval was determined for each significant SNP by investigating the LD between the significant SNP and the flanking SNPs. On each side of the significant SNP, the nearest genomic position where the LD dropped below 0.3 was defined as the boundary of the interval. Next, the defined intervals were merged into independent hQTL based on the following criteria: (1) Two intervals were merged if they overlapped with each other; (2) Two non-overlapping intervals were merged if the distance between the two peak SNPs in the intervals was less than 10 Mb (determined by the average decay of LD across genome) and the average LD between all SNPs in the two intervals was higher than 0.3. The procedure was repeated until no more intervals could be merged, and the resulting intervals were treated as independent hQTL. In each hQTL, the most significant SNP was referred as the lead SNP.

The PVE of each hQTL was estimated by the following formula:

$$\text{PVE}_i = \frac{\Delta_i \hat{\sigma}^2_{h_i}}{\Delta_i \hat{\sigma}^2_{h_i} + \Delta_{\text{D}} \hat{\sigma}^2_{\text{D}} + \Delta_{\text{AA}} \hat{\sigma}^2_{AA} + \Delta_{\text{AD}} \hat{\sigma}^2_{AD} + \Delta_{\text{DD}} \hat{\sigma}^2_{DD} + \hat{\sigma}^2_e}, \tag{20}$$

where the variance components were estimated by fitting the lead SNP of the hQTL in model (8), $\Delta_* = \text{mean}(\text{diag}(\mathbf{K}_*)) - \text{mean}(\mathbf{K}_*)$ (* denotes D, AA, AD or DD) and $\Delta_i = \text{mean}(\text{diag}(\mathbf{H}_i)) - \text{mean}(\mathbf{H}_i)$. Note that multiplying the variance components with the parameter $\Delta$ yields the variance explained by the corresponding genetic values[59].

In additional to the test for heterotic effects, the dominance effects of all markers were tested using model (10) in the integrated panel. The same procedure as described above was used to merge significant SNPs into dQTL. Moreover, for each hQTL in the integrated panel, the digenic epistatic interaction effects between the SNPs in the hQTL and all SNPs across genome were tested using models (11)-(14).

## Simulation study

A simulation study was performed to compare the performance of hQTL-ODS and hQTL-MSS in terms of statistical power and FPR. The simulation was based on the genomic data of Exp I. To reduce the computational load, we randomly sampled a subset of individuals and markers. As a result, the data used for simulation consisted of 90 parental lines (75 females and 15 males) and their 1000 hybrid progenies with 5000 SNPs.

Five different scenarios were considered (Supplementary Data 2). In each scenario, one marker was randomly sampled as hQTL and its PVE was fixed as 2.5%. The heritability was fixed as 0.5. In Scenario 1, the heterotic effect was contributed by cumulative small epistatic effects. One marker from each of the remaining 20 chromosomes (excluding the one on which the hQTL was located) was randomly sampled. The 20 markers interacted with the hQTL, and the pattern of interaction (additive-by-additive, additive-by-dominance, dominance-by-additive, and dominance-by-dominance) was randomly assigned. The PVE for each of the 20 epistatic interaction effects was fixed as 0.5%. Scenario 2 is similar to Scenario 1, but only five markers interacted with the hQTL and the PVE for each of the five epistatic effects was 2%. In Scenario 3, the significant heterotic effect was due to cumulative small dominance and epistatic effects. Thus, the only difference between Scenario 3 and 1 is that one epistatic effect was replaced by the dominance effect of the hQTL itself. Thus, the PVE for the dominance effect of the hQTL and for each of the 19 epistatic interaction effects was 0.5%. Scenario 4 is similar to Scenario 3, but only four markers interacted with the hQTL. The PVE for the dominance effect of the hQTL and for each of the four epistatic effects was 2%. In Scenario 5, the significant heterotic effect was solely due to the dominance effect of the hQTL with 2.5% PVE. Therefore, no epistatic interaction effect was simulated in this scenario.

The simulated phenotypic data (MPH values of 1000 hybrids) were produced by the following steps:

1) Simulating the genetic background effect. Assuming that $\tilde{\sigma}^2_D = \tilde{\sigma}^2_{AA} = \tilde{\sigma}^2_{AD} = \tilde{\sigma}^2_{DD} = 1$, we randomly sampled vectors $\tilde{\mathbf{g}}_D \sim N(0, \mathbf{K}_D \tilde{\sigma}^2_D)$, $\tilde{\mathbf{g}}_{AA} \sim N(0, \mathbf{K}_{AA} \tilde{\sigma}^2_{AA})$, $\tilde{\mathbf{g}}_{AD} \sim N(0, \mathbf{K}_{AD} \tilde{\sigma}^2_{AD})$ and $\tilde{\mathbf{g}}_{DD} \sim N(0, \mathbf{K}_{DD} \tilde{\sigma}^2_{DD})$, where $\mathbf{K}_D$, $\mathbf{K}_{AA}$, $\mathbf{K}_{AD}$ and $\mathbf{K}_{DD}$ were defined previously (after Eq. 7). Then, the vector of genetic background effects was $\tilde{\mathbf{u}} = \tilde{\mathbf{g}}_D + \tilde{\mathbf{g}}_{AA} + \tilde{\mathbf{g}}_{AD} + \tilde{\mathbf{g}}_{DD}$. Since the heritability was fixed as 0.5 and the PVE of hQTL was 2.5%, we knew that the PVE of the genetic background effects should be 47.5%. Thus, the total phenotypic variance $V_t$ was calculated as $V_t = S(\tilde{\mathbf{u}})/0.475$, where $S(\cdot)$ denote the sample variance.

2) Simulating the hQTL effect $\mathbf{h}_i$ using the following formula (reformulated from Eq. 4):

$$\mathbf{h}_i = \mathbf{Tl}_i s_c d_i + \frac{1}{2}\left[\sum_{j \in Q_{\text{aa}}} \mathbf{T}\left(\mathbf{m}_i \circ \mathbf{m}_j\right) aa_{ij} + \sum_{j \in Q_{\text{ad}}} \mathbf{T}\left(\mathbf{m}_i \circ \mathbf{l}_j\right) ad_{ij} + \sum_{j \in Q_{\text{da}}} \mathbf{T}\left(\mathbf{l}_i \circ \mathbf{m}_j\right) da_{ij} + \sum_{j \in Q_{\text{dd}}} \mathbf{T}\left(\mathbf{l}_i \circ \mathbf{l}_j\right) dd_{ij}\right], \tag{21}$$

where $s_c = 0$ for scenarios 1 and 2, and $s_c = 1$ for scenarios 3, 4 and 5; $Q_{\text{aa}}$, $Q_{\text{ad}}$, $Q_{\text{da}}$, and $Q_{\text{dd}}$ are the subsets of markers that were sampled to interact with the hQTL with the assigned pattern. Note that each of them could be the empty set, depending on the scenario and the result of random assignment. For example, all of them were empty sets in scenario 5 because no epistatic effects were simulated. To simplify the description, we denote by $x$ a component effect (i.e., $x$ is $d_i$, $aa_{ij}$, $ad_{ij}$, $da_{ij}$ or $dd_{ij}$), its coefficient in Eq. (21) by $\mathbf{c}_x$ (e.g., $\mathbf{c}_x = \frac{1}{2}\mathbf{T}(\mathbf{m}_i \circ \mathbf{m}_j)$ for $x = aa_{ij}$), and its PVE by $P_x$. Thus, $P_x = 0.5\%$ in scenarios 1 and 3, $P_x = 2\%$ in scenarios 2 and 4, and $P_x = 2.5\%$ in scenario 5. Then, the variance explained by $x$ is $\text{var}(\mathbf{c}_x x) = \text{var}(\mathbf{c}_x)x^2$, which should be $V_t \cdot P_x$. Thus, the effect size $x$ was calculated as $x = \sqrt{V_t \cdot P_x / \text{var}(\mathbf{c}_x)}$. After all component effects were computed, they were summed up to produce $\mathbf{h}_i$ by Eq. (21). Then, the PVE of $\mathbf{h}_i$ is $S(\mathbf{h}_i)/V_t$, but this value is not

guaranteed to be the predetermined value (i.e., 2.5%), nor is it necessarily equal to the sum of all $P_x$, because there exists covariance between the coefficients $\mathbf{c}_x$. Practically, we changed the covariance structure by repeating the random sampling of the markers that interact with the hQTL until $S(\mathbf{h}_i)/V_t$ equaled 2.5%.

3) Simulating the residual effect. Since the heritability was fixed as 0.5, the residual variance is $\widetilde{\sigma}_\varepsilon^2 = 0.5V_t$. Thus, we randomly sampled $\widetilde{\varepsilon} \sim N(0, \mathbf{TT}'\widetilde{\sigma}_\varepsilon^2)$.

4) Generating the simulated MPH values $\widetilde{\mathbf{y}}$ according to Eq. (8), i.e., $\widetilde{\mathbf{y}} = \mathbf{h}_i + \widetilde{\mathbf{u}} + \widetilde{\varepsilon}$. Note that for simplicity, we didn't simulate any covariate effects.

We applied hQTL-ODS and hQTL-MSS to each simulated data set and compared the results. Since only one hQTL was simulated in each data set, the statistical power in each scenario was estimated as the proportion of data sets in which the marker simulated as hQTL was successfully detected. The FPR was estimated as the average proportion of null markers (i.e., 4999 markers that were not simulated as hQTL) that were incorrectly identified as significant across 100 simulations. Note that we used simple criteria based on markers instead of regions, as LD between markers was in most cases not high, given that only 5000 markers were used for simulation.

We investigated the association between the power of detecting hQTL and the heterozygosity. In each scenario, the 100 simulated hQTL were classified into two categories according to whether the heterozygosity was above 0.5 or not. In each category, they were further divided into two classes, depending on whether they were detected by hQTL-ODS or not. Then, Fisher's exact test was performed to assess the significance of association.

### Reporting summary
Further information on research design is available in the Nature Portfolio Reporting Summary linked to this article.

### Data availability
Raw sequence data collected in this study have been deposited at the European Nucleotide Archive under Project PRJEB48738 (https://www.ebi.ac.uk/ena/browser/view/PRJEB48738, parental lines from Exp I, released in a previous study[60]) and PRJEB82869 (https://www.ebi.ac.uk/ena/browser/view/PRJEB82869, parental lines from Exp II and III). VCF files for whole WGS data are available from European Variation Archive under project PRJEB87554. The phenotypic data of all parents and hybrids, as well as the simulated data sets are provided in GitHub (https://github.com/Ligl0226/hQTL-ODS/). The genotypic and phenotypic data of the published maize dataset[8] used in this study were accessed via the CNGBdb FTP public repository (https://ftp.cngb.org/pub/CNSA/data3/CNP0001565/zeamap/99_MaizegoResources/01_CUBIC_related/). Source data are provided with this paper (https://doi.org/10.6084/m9.figshare.30258319).

### Code availability
The hQTL-ODS model was implemented as easy-to-use R functions[61] available at https://github.com/Ligl0226/hQTL-ODS/ and https://doi.org/10.5281/zenodo.17243742.

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

## Acknowledgements

This study was partly supported by the German Research Foundation (DFG) within the Project "Developing a statistical genomics toolbox to decipher the genetic architecture of heterosis using whole-genome sequencing data" (grant number: 540803247). G.L. was financially and logistically supported by a Sino-German (CSC-DAAD) postdoc scholarship program with China Scholarship Council (CSC), Grant no. 202006350258. FAIRification of the data was supported in the frame of the NFDI consortium FAIRagro (www.fairagro.net). We gratefully acknowledge the financial support of DFG (grant number 501899475).

## Author contributions

Y.J. and J.C.R. conceived the study. Y.J. developed the hQTL-ODS model, G.L. implemented the model in R environment and analyzed the data. R.H.S. and Y.Z. provided helpful suggestions at various stages of the study. All authors wrote the manuscript.

## Funding

## Competing interests

The authors declare no competing interests.
