## [Transparent Peer Review file · Nature Communications]

Powerful one-dimensional scan to detect heterotic quantitative trait loci

Corresponding Author: Dr Yong Jiang

Version 0:

Reviewer comments:

Reviewer #1

(Remarks to the Author)

The manuscript introduces a new algorithm, hQTL-ODS, for identifying heterotic QTL, building upon the previously published hQTL-MSS developed by the same group. The innovation for hQTL-ODS lies in integrating all heterotic effects (dominance and digenic epistatic effects) into one term (h^i) so the effect of each SNP can be tested in one model. In contrast, hQTL-MSS tests each SNP with five separately models. The empirical results using wheat datasets convincingly support the effectiveness of hQTL-ODS, and several biologically interesting QTL were highlighted.

The formulation of h^i is particularly clever. I am impressed by the extensive effort for the mathematic proof. and I appreciate the authors' effort in providing rigorous mathematical justification. Because of the importance of heterosis for agricultural production, from the perspective of algorithm development, I believe this work is suitable for publication in Nature Communications. My main suggestions concern the biological interpretation and experimental design.

1. Integration of Multiple Experiments (I, II, III) into one. The rationale for combining these three loosely connected experiments is unclear. Although the merged dataset increases sample size significantly (from ~1,000 to ~4,000 hybrids), the experiments were conducted in different years and settings. This introduces uncertainty in estimating mid-parent heterosis (MPH). While Table S5 shows high correlations among environments for a limited number of overlapping accessions, this does not necessarily guarantee comparable raw phenotypic scales, which may influence MPH estimates. Notably, the major signal on chromosome 6B (Lines 204–205) was not detected in Experiment III, hinting at heterogeneity among experiments. In my opinion, Fig. 4f and Extended Fig. 2f—comparing significant SNPs across datasets—are tangential to the central message. Similar insights from Figs. 4e, 4g, and 5 could have been demonstrated using a single representative experiment. Combining all experiments may also obscure interpretation of “common hQTL” (Lines 194–195), since many of these loci were already shared between individual and integrated analyses. In other words, I feel integrated dataset brings more troubles than benefits.

2. Testing hQTL-ODS Beyond Wheat. Given that hybrid wheat is not widely adopted in current production, it would strengthen the generality of hQTL-ODS to test it in other major hybrid crops such as maize or rice, provided high-quality, open-access datasets are available.

3. Clarifying the Genetic Architecture of MPH. As noted in the group's earlier work (Ref 19), MPH is a derived trait influenced not only by the hybrid genotype, but also by the genetic composition of its parents. For instance, hybrids F14 and F23 in the Supplementary Note of Ref 19 share the same genotype and genotypic value but have different MPH (and h^i). Suppose two populations are developed from parents with different genotype compositions—one composed of 40% R00, 40% R22, 10% R02, and 10% R20, and another with 40% R02, 40% R20, 10% R00, and 10% R22. Would the same QTL still be detectable from MPH in both populations? Elaborating on this would greatly enhance biological interpretation of the method.

A minor point is related to Line 181 and Fig. 5e. Because it has been noticed and mentioned, it is probably better to explain the reason for these SNPs were detected.

I struggled for a while with understanding Equation 4 (Line 308) for h^i , also shown in Fig. 1. If the dimensions of h^i as $n \times 1$ and T as $n \times (n+r)$, the term in the bracket must have a dimension of $(n+r) \times 1$. However, the unknown effects (a_{ij}) and others differ between SNP pairs, implying they cannot be simplified into a one vector across all SNP pairs. Line 349-352 resolve this confusion. I suggest moving these Lines directly after Equation 4. Additional, consider updating the representation for h^i in Fig. 1 to minimize confusions.

(Remarks on code availability)

Please implement a data integrity check to ensure that all parental lines referenced in hybrids (e.g., “ZZ_x_YY”) are present in both phenotype and genotype files. This would help avoid cases where a parent is

missing or misrecorded (e.g., "Y" instead of "YY"). Such case will impact the Tmat and MPH calculation.

The second is in the R01.ODS_code_domoo.R: KinMatlist\$K_da[1:5,1:5] reported NULL. This appears to be a typo for K_aa?

Seems the implementation of Model 10 to 14 that test individual heterotic effect component of detected hQTL has not been included in the current version. If so, please update.

I appreciate that the MPH implementation is also released.

Reviewer #2

(Remarks to the Author)

The authors have developed new methodology and software for a GWAS test of non-additive genetic variance. The novelty of the approach lies in using a single test for the combined contribution of dominance and all forms of digenic epistasis. Compared to the existing paradigm, in which each epistatic interaction is tested in isolation, the proposed method was faster and had higher power in the simulation. The expertise of the authors in statistical genetics is evident.

Major comments

L23: In the abstract, the authors state "Our findings offered valuable insights for improving breeding strategies." I consider this unsupported, as I did not see any development of this idea. I think it is fair to claim this research can help assess the relative importance of dominance vs. epistasis to heterosis, but I'm not sure it has major implications for breeding strategy.

Other comments

L323: I don't think it is correct to call "a" an additive effect; it is half the difference between the homozygotes. This follows standard nomenclature wherein additive effects are based on a regression on allele dosage (and depend on the d parameter).

L339: The genetic background effect for dominance was assumed to have zero mean, but this implies no overall heterosis (Xiang et al. 2016; <https://doi.org/10.1186/s12711-016-0271-4>). Directional dominance models include a covariate for the overall heterozygosity. Was this intentional, to avoid including heterosis in the baseline model? Some mention in the text would be helpful. The same reasoning may apply to dominance epistatic terms.

L349: The multivariate normal heterotic effect, h, is based on a vector of i.i.d. random effects (gamma) that include the dominance and all epistatic interactions with the marker. This contrasts with the baseline model, where each type of effect (d, aa, ad, dd) has its own variance component. The simulations appear to show the method is robust to different genetic architectures, but this model assumption could be highlighted.

L522 The variance of a multivariate normal random effect in the PVE equation is not just the variance component. As shown by Legarra (2016), <http://dx.doi.org/10.1016/j.tpb.2015.08.005>, the VC should be multiplied by $\text{mean}(\text{diag}(K)) - \text{mean}(K)$. By construction, $\text{mean}(\text{diag}(K))=1$ for your K matrices, but the Kd matrix (and perhaps others) does not have zero mean.

(Remarks on code availability)

Version 1:

Reviewer comments:

Reviewer #1

(Remarks to the Author)

Thanks for the thorough responses and revisions. I have no further comments.

(Remarks on code availability)

I have two suggestions

1. During my tests, some SNPs had "EM step failed to improve likelihood (this should not happen)" at "log-likelihood value evaluated from the full model" step. If this is expected, let users aware that this warning is normal (explaining the potential reasons will be better) in the GitHub Readme.

2. Maybe use a much smaller demo data so the whole process can be quickly gone through. I modified the code to only test the first 10 SNPs to the hQTL_Comp_d <- hQTL_CompEffstep in my laptop.

Reviewer #2

(Remarks to the Author)

I think the authors have successfully addressed the reviewer comments and improved the manuscript.

(Remarks on code availability)

Response to the reviewers

We would like to thank the two reviewers for their valuable suggestions, which have been carefully addressed in our revised manuscript. The following is our point-by-point response. Revised texts in the manuscript were highlighted in red color, and the line numbers were indicated in the relevant part of our response.

Reviewer #1 (Remarks to the Author):

The manuscript introduces a new algorithm, hQTL-ODS, for identifying heterotic QTL, building upon the previously published hQTL-MSS developed by the same group. The innovation for hQTL-ODS lies in integrating all heterotic effects (dominance and digenic epistatic effects) into one term (h_i) so the effect of each SNP can be tested in one model. In contrast, hQTL-MSS tests each SNP with five separately models. The empirical results using wheat datasets convincingly support the effectiveness of hQTL-ODS, and several biologically interesting QTL were highlighted.

The formulation of h_i is particularly clever. I am impressed by the extensive effort for the mathematic proof. and I appreciate the authors' effort in providing rigorous mathematical justification. Because of the importance of heterosis for agricultural production, from the perspective of algorithm development, I believe this work is suitable for publication in Nature Communications. My main suggestions concern the biological interpretation and experimental design.

Thank you for the positive feedback on the manuscript. We have addressed the comments regarding biological interpretation and experimental design below.

1. Integration of Multiple Experiments (I, II, III) into one. The rationale for combining these three loosely connected experiments is unclear. Although the merged dataset increases sample size significantly (from ~1,000 to ~4,000 hybrids), the experiments were conducted in different years and settings. This introduces uncertainty in estimating mid-parent heterosis (MPH). While Table S5 shows high correlations among environments for a limited number of overlapping accessions, this does not necessarily guarantee comparable raw phenotypic scales, which may influence MPH estimates. Notably, the major signal on chromosome 6B (Lines 204–205) was not detected in Experiment III, hinting at heterogeneity among experiments. In my opinion, Fig. 4f and Extended Fig. 2f—comparing significant SNPs across datasets—are tangential to the central message. Similar insights from Figs. 4e, 4g, and 5 could have been demonstrated using a single representative experiment. Combining all experiments may also obscure interpretation of “common hQTL” (Lines 194–195), since many of these loci were already shared between individual and integrated analyses. In other words, I feel integrated dataset brings more troubles than benefits.

Thank you very much for this insightful comment. We agree with the reviewer that several of our manuscript's key conclusions could have been drawn from the individual datasets' results. However, we believe that the integrated dataset offers advantages despite presenting some challenges. The reasons are as follows:

Studies have examined the potential benefits and challenges of integrating the data from the three experiments regarding hybrid performance (Zhao et al., 2021, Lell et al., 2024). These studies suggest that combining experiments results in greater precision in genomic prediction and genome-wide association

studies (GWAS). Therefore, it is tempting to speculate that the same is true for dissecting the genetic basis of MPH. In our study, although only a limited number of genotypes overlap across experiments, the parental lines in the three experiments were all elite lines from Central Europe, and there is no clear subpopulation structure (Supplementary Fig. 1a). Thus, the three sets of genotypes evaluated in the three experiments can be treated as three different samples from the same population. Since the phenotypic correlations among the three experiments estimated by using the overlapping genotypes were high (Supplementary Table 6), it indicates that the genotype-by-experiment ($G \times \text{Exp}$) interaction for the traits under consideration is not very strong. In our opinion, these results support data integration.

As the reviewer pointed out, the raw phenotypic scales may not be comparable across data sets despite high correlations. Indeed, the observed scale of MPH values for grain yield differs slightly among the three experiments (Supplementary Fig. 1e). If the experimental influence on the hybrids were the same as on the parental lines, it would have been eliminated from the MPH values by definition. Hence, the $G \times \text{Exp}$ interaction may still play a significant role for MPH. Although modelling $G \times \text{Exp}$ for hQTL-ODS is outside the range of the current study, we did consider the influence of $G \times \text{Exp}$ by investigating the stability of heterotic QTL (hQTL) across experiments and (sub)populations. That is, we conducted hQTL analyses in each experiment and compared the results. On the other hand, performing hQTL analysis in the integrated dataset implicitly considers the influence of all experiments. Although the influence might be negative (reducing the QTL detection power), it could be counteracted by the larger sample size. Therefore, we compared the results obtained in the integrated dataset and those from each individual experiment.

Of course, it is difficult to compare significant SNPs or hQTL across datasets. We focused in the comparison on the proportion of “common hQTL” in individual experiments and in the integrated dataset. As the reviewer mentioned, the results clearly indicate heterogeneity among the experiments. It may be impossible to determine why a certain hQTL was present in one dataset but absent in another. For hQTL identified in only one of the three experiments, it is reasonable to assume that they are more reliable if they were also detected in the integrated dataset. Hence, we considered all hQTL identified in two or more datasets (individual experiments or the integrated dataset) as “common hQTL”. The much higher proportion of “common hQTL” in the integrated dataset (74.5%) than in the individual experiments (less than 40%) (lines 206-211) suggests that integrating datasets is a promising approach for detecting reliable QTL.

Summarizing, we hope that we have addressed the challenges of integrating data across experiments. We have retained the results of the integrated panel in the revised manuscript. Further in-depth analyses for $G \times \text{Exp}$ may be warranted, but it is beyond the scope of our methodological study.

2. Testing hQTL-ODS Beyond Wheat. Given that hybrid wheat is not widely adopted in current production, it would strengthen the generality of hQTL-ODS to test it in other major hybrid crops such as maize or rice, provided high-quality, open-access datasets are available.

Thank you for highlighting the potential of the hQTL-ODS model for other important crops, thereby strengthening the validity of our statements. We used a publicly available maize dataset (Xiao et al. 2021) consisting of 6,210 single-cross hybrids from partial crosses of 207 with 30 parental inbred lines. Data on grain yield was unavailable, but data on the ear weight (EW) was available. Our analyses demonstrated the potential of the hQTL-ODS model: While we did not detect any significant dominance effects, we identified an interesting hQTL on chromosome 3, in which a SNP (chr3.s_159023566, $P = 5.63 \times 10^{-6}$) mapped to MADS-box transcription factor *ZmMADS69* and the peak SNP (chr3.s_157746554, $P = 2.56 \times$

10^{-8}) was within a distance of 1.28Mb. *ZmMADS69* functions as a flowering activator but due to pleiotropy also affects other traits of agronomic importance such as ear size (Liang et al. 2019).

Xiao et al. (2021) performed GWAS separately in 30 F_1 populations using a different approach. Each of the 30 F_1 populations consists of 1,428 hybrids, but only two were fully evaluated in field trials. For the remaining 28 F_1 populations, phenotypic data for 207 hybrids were available and the performance of the remaining 1,221 hybrids was predicted by using genomic data. *ZmMADS69* was identified for days to tasseling (DTT) and for plant height (PH), but not for EW. However, supplementary data (Xiao et al. 2021: Additional file 5) shows that QTL for EW at positions near *ZmMADS69* were detected in about half of the F_1 populations (in most cases at the lower threshold $P < 10^{-4}$). Additionally, merged QTL very close to *ZmMADS69* were listed for PH and EW with comparatively large epistatic effects (see Xiao et al. 2021: Additional file 6). Thus, despite using different methods and populations, our results for *ZmMADS69* are consistent with the findings of Xiao et al. (2021).

We have integrated our results into the revised manuscript (lines 241-255).

3. Clarifying the Genetic Architecture of MPH. As noted in the group's earlier work (Ref 19), MPH is a derived trait influenced not only by the hybrid genotype, but also by the genetic composition of its parents. For instance, hybrids F14 and F23 in the Supplementary Note of Ref 19 share the same genotype and genotypic value but have different MPH (and hi). Suppose two populations are developed from parents with different genotype compositions—one composed of 40% R00, 40% R22, 10% R02, and 10% R20, and another with 40% R02, 40% R20, 10% R00, and 10% R22. Would the same QTL still be detectable from MPH in both populations? Elaborating on this would greatly enhance biological interpretation of the method.

This is an interesting question. In fact, even in the same population, the composition of the parental genotypes (R00, R22, R02 and R20) are different for distinct markers. At a single locus, the proportion of R02 and R20 together equals the level of heterozygosity. Thus, if the hQTL effect is mainly contributed by the dominance effect of the locus itself, the QTL detection power would be associated with the heterozygosity – just like the relationship between the detection power for additive effects and the minor allele frequency in normal GWAS. However, if cumulative epistatic effects play an important role, the picture would be much more complex and difficult to anticipate in theory. Many factors, such as the type of epistasis (additive-by-additive, additive-by-dominance and dominance-by-dominance), the number of loci interacting with the hQTL and the sizes of epistatic effects, could affect the power of detection.

Therefore, we followed the suggestion and elaborated the power of QTL detection for different compositions of parental genotypes in our simulation study. In each of the five simulated scenarios, we classified the simulated hQTL (one in each of the 100 runs, so in total 100 hQTL) into two categories according to whether the heterozygosity is above 0.5 or not. The hQTL in each category was further divided into two classes, depending on whether they were detected by hQTL-ODS or not. Then, we performed Fisher's exact test to investigate whether the power of detection is associated with the heterozygosity. As expected, the result was very significant ($P < 0.0001$) for Scenario 5 in which the hQTL effect was solely contributed by the dominance effect. In Scenarios 3 and 4 where both dominance and epistatic effects contributed to the hQTL effect, although the power of detection was still higher in the category of heterozygosity above 0.5, the association was not significant ($P > 0.05$). For scenarios in which epistatic effects shaped out the hQTL, the association became even weaker in Scenario 2 and an opposite trend

was observed (non-significant) in Scenario 1. These results are in line with expectation and were added to the revised manuscript (lines 137-147).

A minor point is related to Line 181 and Fig. 5e. Because it has been noticed and mentioned, it is probably better to explain the reason for these SNPs were detected.

The sentence was modified as “Notably, 15 of the 105 hQTL (13.9%) showed neither significant dominance nor epistatic interactions effects with other loci, indicating that cumulative small epistatic effects composed these hQTL” (lines 192-193).

I struggled for a while with understanding Equation 4 (Line 308) for h_i , also shown in Fig. 1. If the dimensions of h_i as $n \times 1$ and T as $n \times (n + r)$, the term in the bracket must have a dimension of $(n + r) \times 1$. However, the unknown effects (a_{ij} and others differ between SNP pairs, implying they cannot be simplified into a one vector across all SNP pairs. Line 349-352 resolve this confusion. I suggest moving these Lines directly after Equation 4. Additionally, consider updating the representation for h_i in Fig. 1 to minimize confusions.

We followed the suggestion and revised the explanations for Equation (4) (lines 349-352). For Fig. 1, there would be too many math equations if we update it accordingly. Since the figure just provides an overview of the current method compared with the existing one, instead of explaining the approach in details, we would prefer not to introduce these changes.

Reviewer #1 (Remarks on code availability):

Please implement a data integrity check to ensure that all parental lines referenced in hybrids (e.g., "ZZ_x_YY") are present in both phenotype and genotype files. This would help avoid cases where a parent is missing or misrecorded (e.g., "Y" instead of "YY"). Such case will impact the Tmat and MPH calculation.

We followed the suggestion and added a data integrity check in the example code (R01.ODS_code_domoo.R) to ensure that all parental lines referenced in the hybrids are present in both the phenotype and genotype files. The updated code has been uploaded to the corresponding GitHub repository (https://github.com/Ligl0226/hQTL-ODS/blob/main/R01.ODS_code_domoo.R).

The second is in the R01.ODS_code_domoo.R: `KinMatlist$K_da[1:5,1:5]` reported NULL. This appears to be a typo for `K_aa`?

Yes, thank you very much for spotting this error. It was indeed a typo and the correct term should be `K_aa`. We have corrected it in the updated version of the code (R01.ODS_code_domoo.R).

Seems the implementation of Model 10 to 14 that test individual heterotic effect component of detected hQTL has not been included in the current version. If so, please update.

Yes, we followed the suggestion and added an R function for testing the individual component effects of the detected hQTL in the GitHub source code folder (https://github.com/Ligl0226/hQTL-ODS/blob/main/R/hQTL_component_effects.R). This function implements Models 10 to 14 as described

in the manuscript, and its usage is clearly demonstrated in the provided R example script (R01.ODS_code_domoo.R).

I appreciate that the MPH implementation is also released.

We appreciate reviewer #1 taking such a detailed look at the implementation. Thank you.

Reviewer #2 (Remarks to the Author):

The authors have developed new methodology and software for a GWAS test of non-additive genetic variance. The novelty of the approach lies in using a single test for the combined contribution of dominance and all forms of digenic epistasis. Compared to the existing paradigm, in which each epistatic interaction is tested in isolation, the proposed method was faster and had higher power in the simulation. The expertise of the authors in statistical genetics is evident.

Thank you for the positive feedback.

Major comments

L23: In the abstract, the authors state “Our findings offered valuable insights for improving breeding strategies.” I consider this unsupported, as I did not see any development of this idea. I think it is fair to claim this research can help assess the relative importance of dominance vs. epistasis to heterosis, but I’m not sure it has major implications for breeding strategy.

We agreed with the reviewer. This sentence was removed from the abstract.

Other comments

L323: I don’t think it is correct to call “a” an additive effect; it is half the difference between the homozygotes. This follows standard nomenclature wherein additive effects are based on a regression on allele dosage (and depend on the d parameter).

Yes, we are aware that it is strongly discouraged to call “a” additive effect in quantitative genetics and this is certainly reasonable. We decided to call “a” and “d” the “coded genotypic effects” following Bernardo (2010) (lines 365-367, line 440).

L339: The genetic background effect for dominance was assumed to have zero mean, but this implies no overall heterosis (Xiang et al. 2016; <https://doi.org/10.1186/s12711-016-0271-4>). Directional dominance models include a covariate for the overall heterozygosity. Was this intentional, to avoid including heterosis in the baseline model? Some mention in the text would be helpful. The same reasoning may apply to dominance epistatic terms.

Thank you for raising this interesting point. It is true that the mean heterosis is assumed to be zero according to our null model (Eq. 1) and the final model for testing the heterotic effect (Eq. 2), provided that there are no covariate effects (α). In our settings, the null model for heterosis is linearly transformed from the null model for the original trait (Eq. 5), and it is a common assumption that in such a model all background effects (a, d, aa, ad, dd) have zero means (e.g. Xu et al. 2013). When QTL effects are involved

in the model, they are usually (and reasonably) assumed to be fixed effects in almost all GWAS models. Following this convention, this assumption should also apply to the heterotic effect h . Hence, theoretically, the final genetic model involving all hQTL effects is

$$\mathbf{y}_{MPH} = \mathbf{X}\boldsymbol{\alpha} + \sum_{i=1}^q \mathbf{h}_i + \mathbf{g}_D + \mathbf{g}_{AA} + \mathbf{g}_{AD} + \mathbf{g}_{DD} + \boldsymbol{\varepsilon},$$

where q is the number of hQTL. Then, the mean of \mathbf{y}_{MPH} should be $\mathbf{X}\boldsymbol{\alpha} + \sum_{i=1}^q \mathbf{h}_i$, which is not zero even if $\boldsymbol{\alpha} = \mathbf{0}$.

However, if we keep the assumption that \mathbf{h}_i is a fixed effect, which means that all component effects have to be assumed to be fixed, then \mathbf{h}_i is a combination of 4p-3 fixed effects. As this number exceeds the number of observations, it would be difficult to perform a proper statistical test for \mathbf{h}_i . Therefore, we switched from fixed to random effects, making the likelihood-ratio test possible. Although the model seems to be mis-specified for traits with a non-zero mean heterosis, the results in the simulation study suggested that the testing procedure is robust in many different scenarios. We added a short paragraph in Discussion to clarify this issue (lines 277-286).

In Xiang et al. (2016), a directional dominance model (the dominance effects have non-zero mean values) was introduced, which is equivalent to including a covariate for the overall heterozygosity. This is indeed a very interesting approach to deal with the case in which the average heterosis is not zero. This could be considered in further development of our hQTL-ODS model. We would like to thank the reviewers again for pointing out this reference and we highlighted this in the revised Discussion (lines 286-290).

L349: The multivariate normal heterotic effect, h , is based on a vector of i.i.d. random effects (γ) that include the dominance and all epistatic interactions with the marker. This contrasts with the baseline model, where each type of effect (d , aa , ad , dd) has its own variance component. The simulations appear to show the method is robust to different genetic architectures, but this model assumption could be highlighted.

Indeed, the final model for testing the heterotic effect h relies on a different assumption on the variance of the component genetic effects, compared with the baseline model. With this simplified assumption, the number of random terms in the final model is five (excluding the residuals). Otherwise, we would have to split the heterotic effect into four different parts (d , the combination of aa , ad and dd) and as a consequence, the number of random terms would be eight. In our opinion, this would very likely make the model over-parametrized. Hence, we decided to simplify the assumption in the final model, and as the reviewer mentioned, it seems that the method is robust in distinct scenarios of simulation. We added an explanation in the Methods sections to clarify this point (lines 399-406).

L522 The variance of a multivariate normal random effect in the PVE equation is not just the variance component. As shown by Legarra (2016), <http://dx.doi.org/10.1016/j.tpb.2015.08.005>, the VC should be multiplied by $\text{mean}(\text{diag}(K)) - \text{mean}(K)$. By construction, $\text{mean}(\text{diag}(K))=1$ for your K matrices, but the Kd matrix (and perhaps others) does not have zero mean.

Thank you very much for pointing out this important issue. We carefully read the publication (Legarra 2016) and agreed with the reviewer that in our case, the variance components are not the same as the variance

explained by the genetic values, because the mean of all entries in our kinship matrix (K_d , K_{aa} , K_{ad} or K_{dd}) as defined in Methods is not necessarily zero. Therefore, we followed the suggestion and re-estimated the PVE of the hQTL. Surprisingly, the difference between the old and the new estimates was very small. After double checking the computer codes and the descriptions in the text, we realized that the definition of the kinship matrices had not been correctly described. Actually, we had adjusted the coding matrix following Vitezica et al. (2013) when calculating the kinship matrices so that the variance components match the classical concept of dominance and epistatic variance (see <https://github.com/Ligl0226/hQTL-ODS/blob/main/R/CalKinMat.R>). Since the hybrid wheat populations considered in our study were derived from (partial) factorial design, the allele frequencies of the markers are close to Hardy-Weinberg equilibrium. Thus, according to Legarra (2016), the mean of all entries of the kinship matrices calculated for our wheat data sets should be close to zero. This was indeed the case.

We corrected the descriptions in Methods (lines 383-392, lines 576-580) and updated the PVE estimates (Supplementary Table 8, 10). Meanwhile, we also updated the PVE estimation code of the relevant functions in the GitHub repository (https://github.com/Ligl0226/hQTL-ODS/blob/main/R/MPH_hQTL_ODS.R and [BPH_hQTL_ODS.R](https://github.com/Ligl0226/hQTL-ODS/blob/main/R/BPH_hQTL_ODS.R)). We would like to thank the reviewer again for raising this point and we apologize for having overseen the inconsistency between the text and the implementation.

References

- Legarra, A. (2016). Comparing estimates of genetic variance across different relationship models. *Theoretical population biology*, 107, 26-30.
- Lell, M., Zhao, Y., & Reif, J. C. (2024). Leveraging the potential of big genomic and phenotypic data for genome-wide association mapping in wheat. *The Crop Journal*, 12(3), 803-813.
- Liang, Y., Liu, Q., Wang, X., Huang, C., Xu, G., Hey, S., ... & Tian, F. (2019). ZmMADS69 functions as a flowering activator through the ZmRap2.7-ZCN8 regulatory module and contributes to maize flowering time adaptation. *New Phytologist*, 221(4), 2335-2347.
- Vitezica, Z. G., Varona, L., & Legarra, A. (2013). On the additive and dominant variance and covariance of individuals within the genomic selection scope. *Genetics*, 195(4), 1223-1230.
- Xiang, T., Christensen, O. F., Vitezica, Z. G., & Legarra, A. (2016). Genomic evaluation by including dominance effects and inbreeding depression for purebred and crossbred performance with an application in pigs. *Genetics Selection Evolution*, 48(1), 92.
- Xiao, Y., Jiang, S., Cheng, Q., Wang, X., Yan, J., Zhang, R., ... & Yan, J. (2021). The genetic mechanism of heterosis utilization in maize improvement. *Genome Biology*, 22(1), 148.
- Xu, S. (2013). Mapping quantitative trait loci by controlling polygenic background effects. *Genetics*, 195(4), 1209-1222.
- Zhao, Y., Thorwarth, P., Jiang, Y., Philipp, N., Schulthess, A. W., Gils, M., ... & Reif, J. C. (2021). Unlocking big data doubled the accuracy in predicting the grain yield in hybrid wheat. *Science Advances*, 7(24), eabf9106.

Response to the reviewers

Reviewer #1 (Remarks to the Author):

Thanks for the thorough responses and revisions. I have no further comments.

Reviewer #1 (Remarks on code availability):

I have two suggestions

1. During my tests, some SNPs had "EM step failed to improve likelihood (this should not happen)" at "log-likelihood value evaluated from the full model" step. If this is expected, let users aware that this warning is normal (explaining the potential reasons will be better) in the GitHub README.

Thank you very much for carefully testing the code. In hQTL-ODS pipeline, the likelihood estimation is performed using the `Imm.aireml()` function from the R package `gaston`, and this is where this warning originates. Theoretically, a fundamental property of the EM algorithm is that each iteration should not decrease the likelihood value (monotonic increase or remain unchanged). However, in practice, small decreases in likelihood may occasionally occur due to numerical issues such as floating-point error or a very small tolerance parameter. These decreases are typically negligible, and the iterations continue successfully. Subsequent AI-REML updates or EM corrections generally restore the likelihood trajectory. Therefore, this few warnings can be considered normal and does not affect the final results. We have now added an explanation of this behavior to the GitHub README file to make users aware of the potential causes and to clarify that the warning is not problematic.

2. Maybe use a much smaller demo data so the whole process can be quickly gone through. I modified the code to only test the first 10 SNPs to the `hQTL_Comp_d <- hQTL_CompEffstep` in my laptop.

We followed the suggestion and added an additional mini demo dataset (https://github.com/LigI0226/hQTL-ODS/blob/main/data/PHdata_GNdata_miniDemo.Rdata) to the GitHub repository. This dataset included 31 parental lines and 200 hybrids, together with genotypic data for 5,000 SNPs. With this mini dataset, the entire demo workflow can be completed in approximately 20 minutes.

Reviewer #2 (Remarks to the Author):

I think the authors have successfully addressed the reviewer comments and improved the manuscript.